# *cxcl18b*-defined transitional state-specific nitric oxide drives injury-induced Müller glia cell-cycle re-entry in the zebrafish retina

**Aojun Ye[1,2], Shuguang Yu[3], Meng Du[4], Dongming Zhou[4], Jie He[3]\*, Chang Chen[1,2]\***

[1]National Laboratory of Biomacromolecules, CAS Center for Excellence in Biomacromolecules, Institute of Biophysics, Chinese Academy of Sciences, Beijing, China; [2]University of Chinese Academy of Sciences, Beijing, China; [3]Center for Excellence in Brain Science and Intelligence Technology (Institute of Neuroscience), Chinese Academy of Sciences, Beijing, China; [4]Tianjin Medical University, Tianjin, China

**\*For correspondence:**
jiehe@ion.ac.cn (JH);
changchen@moon.ibp.ac.cn (CC)

**Competing interest:** The authors declare that no competing interests exist.

## eLife Assessment

Following retinal injury, zebrafish Müller glia reenter the cell cycle and generate replacement cells; this potentially **valuable** study proposes that injury induces a *cxcl18b*+ transitional state in Müller cells, which then express nitric oxide, inhibiting Notch signaling and allowing Müller glial cells to reenter the cell cycle. However, the evidence supporting the claims is **incomplete**, and the authors have made interpretations and conclusions that are not supported by the data. Questions of the temporal expression and function of cxcl18b, as well as the source of potential inflammatory cues before cxcl18b expression, remain unanswered and technical limitations and data inconsistencies raise concerns. Using larval animals complicates the analysis since the retina is still forming, and distinguishing between injury-induced regeneration and ongoing development is complex. With more rigorous testing of the signaling pathways proposed and a clear demonstration of their inter-dependence, the link between nitric oxide signaling and Notch activity, particularly, would interest those investigating retinal regeneration.

**Abstract** In lower vertebrates, retinal Müller glia (MG) exhibit a life-long capacity of cell-cycle re-entry to regenerate neurons following the retinal injury. However, the mechanism driving such injury-induced MG cell-cycle re-entry remains incompletely understood. Combining single-cell transcriptomic analysis and in vivo clonal analysis, we identified previously undescribed *cxcl18b*-defined MG transitional states as essential routes toward MG proliferation following green/red cone (G/R cone) ablation. Inflammation blockage abolished the triggering of these transitional states, which expressed the gene modules shared by cells of the ciliary marginal zone (CMZ), where life-long adult neurogenesis takes place. Functional studies of the redox properties of these transitional states further demonstrated the regulatory role of nitric oxide (NO) produced by *Nos2b* in injury-induced MG proliferation. Finally, we developed a viral-based strategy to specifically disrupt *nos2b* in *cxcl18b*-defined MG transitional states and revealed the effect of transitional state-specific NO signaling. Our findings elucidate the precision redox mechanism underlying injury-induced MG cell-cycle re-entry, providing insights into species-specific mechanisms for vertebrate retina regeneration.

## Introduction

Unlike mammalian counterparts, retinal Müller glia (MG) in aquatic and amphibian species can enter injury-induced regeneration program (*Goldman, 2014*; *Lahne et al., 2020b*; *Wan and Goldman, 2016*), which unfolds a set of temporal events from glial reactivation, cell-cycle re-entry, new neuron generation, and circuit integration (*Abraham et al., 2024*; *Powell et al., 2016*). In zebrafish, MG respond to injury by glial reactivation, which refers to the initial response of quiescent MG, characterized by morphological changes and upregulation of reactive markers (*gfap, ascl1a, lin28a*) and activation of signaling pathways such as Notch, Jak/Stat, and Wnt (*Lahne et al., 2020b*; *Pollak et al., 2013*; *Sifuentes et al., 2016*; *Yao et al., 2016*). This is followed by the proliferation phase, in which MG initiate a transcriptional shift toward a progenitor-like state and cell-cycle re-entry. Many genes induced by glial reactivation, such as *lin28a, sox2*, and *mycb/h*, are required for cell-cycle re-entry and subsequent neurogenesis (*Gorsuch et al., 2017*; *Jorstad et al., 2017*; *Lee et al., 2024*; *Ramachandran et al., 2010*). Diverse signaling pathways have been identified to participate in this initial regenerative response (*Campbell et al., 2021*; *Wan and Goldman, 2017*).

In contrast, although forced expression of factors like *Ascl1* or deletion of repressive chromatin regulators can partially reprogram MG into interneuron-like cells in the injured mouse retina (*Hoang et al., 2020*; *Jorstad et al., 2017*), the knowledge of the upstream signals that trigger injury-induced cell-cycle re-entry, a critical step for producing neurogenic progenitors, remains limited (*Lenkowski and Raymond, 2014*; *Vihtelic and Hyde, 2000*; *Wu et al., 2001*). The mechanistic understanding of MG cell-cycle re-entry in the injured zebrafish retina will provide critical knowledge for inspiring new strategies for in vivo reprogramming MG to repair the human retina.

The intrinsic and environmental factors contribute to injury-induced MG proliferation (*Gao et al., 2021*; *Lahne et al., 2020b*; *Xiao et al., 2023*). Recent efforts in single-cell RNA-sequencing (scRNA-seq) have identified *ascl1α, clcf1/crlf1a* (*Boyd et al., 2023*), and *mycb/mych* (*Lee et al., 2024*) as key intrinsic drivers of injury-induced MG proliferation in the zebrafish retina. Also, the elegant study of MG injury responses across multiple species, including zebrafish, chicken, and mice, revealed the *hmga1/yap1* signaling network as a key regulator of MG reactivation and neurogenesis in the regenerating retina (*Hoang et al., 2020*). Interestingly, previous studies further reported that post-injury MG in the zebrafish is capable of re-acquiring the regeneration program that largely recapitulates the embryonic retinal developmental program (*Celotto et al., 2023*; *Hoang et al., 2020*; *Lahne et al., 2020a*; *Lyu et al., 2023*).

In terms of extrinsic cues, two sources, those from damaged neurons and injury-recruited microglia, are considered indispensable for injury-induced MG proliferation (*Leach et al., 2021*; *White et al., 2017*). Previous studies have shown that signals derived from various retinal neuron injury models trigger MG regeneration (*Lyu et al., 2023*). Meanwhile, accumulating evidence increasingly appreciates the role of inflammatory signals derived from injury-recruited microglia, such as *TNFα* (*Conner et al., 2014*; *Nelson et al., 2013*), cytokines IL-1β, and IL-10 (*Lu and Hyde, 2024*). Notably, inflammatory responses are intrinsically associated with redox signaling, which is involved in the regeneration processes in various non-neuronal tissues (*Breus and Dickmeis, 2021*; *Jaeschke, 2000*; *Zhang et al., 2024*). However, the involvement of the redox signaling resulting from microglia-derived inflammatory signals in MG cell-cycle re-entry following retina damage remains to be clarified. Nitric oxide (NO), a redox signal, has participated in regenerating various tissues in zebrafish, including the heart (*Rochon et al., 2020*; *Yu et al., 2024*), spinal cord (*Bradley et al., 2010*), and fin (*Matrone et al., 2021*). Endogenous NO is derived from three forms of NO synthase (NOS) in mammals: neuronal NOS (nNOS), endothelial NOS (eNOS), and inducible NOS (iNOS). The zebrafish comprises *Nos1*, a form of neuronal *Nos*, and *Nos2a/2b*, two inducible *Nos*.

In this study, we performed scRNA-seq to characterize the post-injury MG states in the zebrafish retina following the specific ablation of G/R cone. This comprehensive analysis led us to define a set of previously unknown *cxcl18b*-positive transitional states as an essential route of MG cell-cycle re-entry in response to injury, and the inflammatory response was indispensable for the induction of these transitional states. Intriguingly, we found that these *cxcl18b*-defined transitional states exhibited gene patterns shared by cell states in the ciliary marginal zone (CMZ), favoring the idea that *cxcl18b*-defined MG transitional states might represent the developmental program conserved by neuronal regeneration beyond embryonic development. A remarkable phenotype of enriched redox features in these gene patterns suggested the importance of redox signaling. The subsequent screening of

redox-related genes revealed the essential role of NO, produced by *Nos2b*, in triggering MG cell-cycle re-entry after the retinal injury. Notably, we developed a sophisticated viral-based approach to achieve the *cxcl18b*-defined MG transitional state-specific knockout of *nos2b* successfully and verified the requirement of transitional state-specific NO signaling in injury-induced MG proliferation. These findings provide novel cellular and molecular insights into this species-specific post-injury MG cell-cycle re-entry process, with potential implications for the development of regenerative medicine strategies.

## Results

### Single-cell transcriptome analysis reveals the landscape of injury-induced MG states

Zebrafish MG can respond to retinal injury by cell-cycle re-entry, a critical step evolutionarily absent from their mammalian counterparts but essential for neuron regeneration (*Goldman, 2014*; *Powell et al., 2016*). We created a zebrafish retinal injury model by crossing *Tg(opn1lws2: nfsb-mCherry)^uom3* (referred to as *Tg(lws2: nfsb-mCherry)*) with *Tg(mpeg1: GFP)* fish, in which the bacterial nitroreductase (NTR) enzyme was specifically expressed in G/R cone. We selectively ablated the G/R cone starting at 5 days post-fertilization (dpf) by a subsequent 120 hr of metronidazole (MTZ) exposure (*Curado et al., 2007*; *Curado et al., 2008*; *Figure 1A*, *Figure 1—figure supplement 1A*). G/R cone became significantly reduced in number since 48 hr post-injury (hpi) and was mostly depleted at 120 hpi (*Figure 1B*). Meanwhile, a number of microglia (marked by *Tg(mpeg1: GFP)*) migrated to the outer nuclear layer (ONL) as early as 48 hpi, peaked at 72 hpi, and began to reduce in number at 96 hpi and onward (*Figure 1C*). To confirm the identity of these proliferating cells, we performed BLBP immunostaining and observed that the PCNA⁺ cells were also BLBP⁺ (*Figure 1—figure supplement 1B*), indicating their MG origin. Notably, in response to G/R cone ablation, the proliferative MG population increased starting at 48 hpi, peaked at 72 hpi, and began to decline since 96 hpi and forward (*Figure 1D*). Considering the result that the number of proliferative MG peaked at 72 hpi, we focused on the 72 hpi time point for further exploration of MG proliferative behaviors following G/R cone ablation (*Figure 1E*).

By revisiting previously obtained scRNA-seq data of MG enriched from *Tg(lws2: nfsb-mCherry)* crossed with *Tg(gfap: EGFP)* and *Tg(her4.1: dRFP)* retina before and after G/R cone ablation at 72 hpi (*Krylov et al., 2023*), we selected 5932 and 3999 MG cells and their derived progenies from the uninjured and 72 hpi retina, respectively (details in Materials and methods; *Figure 1—figure supplement 1C*). By clustering these cells, we identified 13 clusters, 8 out of which (Clusters 2, 3, 5, 6, 9, 10, 11, and 12) with an increased proportion in response to the G/R cone ablation (*Figure 1—figure supplement 1D*). Subsequently, we re-clustered cells of these 8 clusters (695 cells of uninjured retinae and 3477 cells of 72 hpi retinae), aggregating into 10 new clusters. After the quality control procedure, we did not consider Clusters 7/8/9 due to their small populations with ribosomal, dendritic, and doublet features, resulting in 7 clusters for further analysis (details in Materials and methods; *Figure 1F*).

We performed the pseudo-time trajectory analysis to reveal the progression of these MG clusters following the cone ablation (*Figure 1G*). Cluster 4 cells were highly expressing genes related to mature MG (*glula*, *slc1a2b*, *apoeb*, and *rlbp1a*) (*Bernardos and Raymond, 2006*; *Raymond et al., 2006*; *Thummel et al., 2008*; *Yurco and Cameron, 2005*), the quiescent state (*cx43*) (*Dermietzel et al., 2000*; *Janssen-Bienhold et al., 1998*), and major MG population marker (*fgf24*) (*Krylov et al., 2023*; *Figure 1—figure supplement 1E*). Furthermore, Cluster 4 began to express *s100α10b* and *gfap*, reactive state markers (*Celotto et al., 2023*; *Hoang et al., 2020*), in the injured retinae, but not in the uninjured retina (*Figure 1—figure supplement 1E*). Thus, we set Cluster 4 as the early transitional MG state.

To examine the transition of these 7 MG clusters, the pseudo-time trajectory showed that the main developmental branch consisted of Cluster 4/1/2 and then became divergent into two sub-branches, including Clusters 0 and 5/3/6 (*Figure 1G*). In contrast to the sub-branch of Cluster 0, the sub-branch of Cluster 5/3/6 was highly expressing proliferative cell markers (*pcna*, *mki67*, and *mcm2*). Within the latter sub-branch, while Clusters 3 and 6 had the highest levels of proliferative cell markers, Cluster 6 began to express neuronal differentiation factors (*otx5*, *crx*, and *pde6gb*) (*Abalo et al., 2020*; *Asaoka et al., 2014*; *Shen and Raymond, 2004*; *Figure 1—figure supplement 1E*). Thus, we identified 6

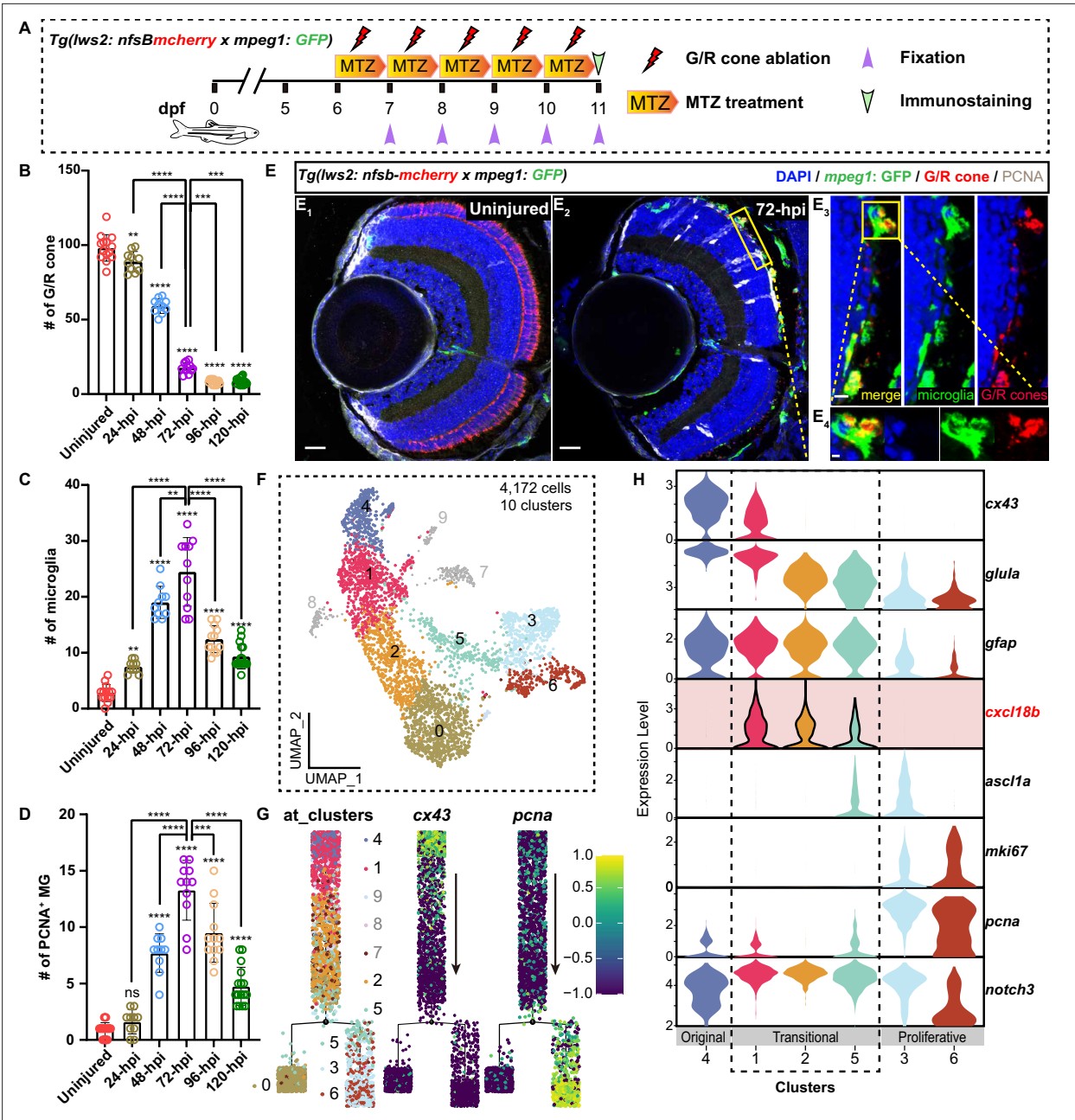

**Figure 1.** Single-cell RNA-sequencing (scRNA-seq) reveals injury-induced *cxcl18b*-defined Müller glia (MG) transitional states. (**A**) Schematic showing the experimental procedure: 5 consecutive days of metronidazole (MTZ) treatment in *Tg(lws2: nfsb-mCherry* x *mpeg1: GFP)* fish to ablate green or red (G/R) cone, starting at 6 days post-fertilization (dpf) and continuing until 11 dpf. The MTZ solution was refreshed every 24 hr, followed by fish fixation for further immunostaining. (**B–D**) Quantitative plots showing the dynamic changes in the number of G/R cone (**B**), recruited microglia (**C**), and proliferative MG (PCNA⁺) (**D**) at different time points after MTZ treatment (uninjured: collected retina number n=14; 24 hr post-injury [hpi]: n=10; 48 hpi: n=10; 72 hpi: n=11; 96 hpi: n=12; 120 hpi: n=16; mean ± SEM; ****p<0.0001, ***p<0.001, **p<0.01, ns, p>0.05; one-way ANOVA followed by Tukey's HSD test). (**E**) Representative images showing microglia recruitment (*mpeg1: GFP*, green), G/R cone ablation (*lws2: nfsb-mCherry*, red), and injury-induced MG proliferation (PCNA, white) in uninjured (**E₁**) and 72 hpi (**E₂**) retinas. The high-magnification images of the boxed area (**E₃– E₄**). Scale bars: 20 μm (**E₁, E₂**), 10 μm (**E₃**), 2 μm (**E₄**). (**F**) The UMAP plot of 4172 MG cells was sorted with an increased proportion in response to the G/R cone ablation. Cells were further aggregated into 10 clusters based on previously published scRNA-seq data (**Krylov et al., 2023**). (**G**) Pseudo-time developmental trajectory of MG states identified by Monocle2 analysis shows a main developmental branch originating from Cluster 4 (*cx43⁺*), which diverges into two sub-branches: Cluster 0 and Clusters 5/3/6 (*pcna⁺*). (**H**) Violin plots showing the expression levels of key genes (*cx43, glula, gfap, cxcl18b, ascl1α, mki67, pcna*, and *notch3*) in the main developmental branch clusters, progressing from the most original MG states (Cluster 4) to transitional MG states (Cluster 1/2/5), and proliferative MG states (Cluster 3/6).

*Figure 1 continued on next page*

*Figure 1 continued*

The online version of this article includes the following source data and figure supplement(s) for figure 1:

**Source data 1.** Quantification of the number of green/red (G/R) cones, recruited microglia, and PCNA$^+$ Müller glia (MG) in the zebrafish retina after injury.

**Figure supplement 1.** Clusters with increased proportion are identified from the single-cell RNA-sequencing (scRNA-seq) data.

major post-injury MG states, from the early transitional state (Cluster 4) to three transitional states (Cluster 1/2/5), to finally two proliferative states (Cluster 3/6).

Remarkably, chemokine (C-X-C motif) ligand 18b (*cxcl18b*), an inflammatory chemokine, was uniquely expressed in three transitional states but largely absent from the early transitional Cluster 4 and two proliferative Cluster 3/6. Specifically, while the first *cxcl18b*$^+$ transitional state (Cluster 1) was expressing *cx43*, a marker for MG quiescence (*Dermietzel et al., 2000*; *Janssen-Bienhold et al., 1998*), the last transitional state (Cluster 5) began to show a weak induction of *ascl1α* (*Ramachandran et al., 2010*). Immediately following this last transitional state, Cluster 3 started with high *ascl1α* expression and entered the proliferative state with the expression of *pcna* and *mik67* (*Figure 1H*). Our analysis highlighted a new set of *cxcl18b*-defined MG transitional states preceding *ascl1α* induction, bridging MG from the most original quiescence state to injury-induced proliferation.

## Clonal analysis reveals injury-induced MG proliferation via *cxcl18b*-defined transitional states associated with inflammation

To directly verify the presence of *cxcl18b*-defined MG transitional states, we first examined the temporal relationship of *cxcl18b* expression and MG proliferation after the cone ablation using in situ hybridization combined with immunostaining of either BLBP (an MG marker) or PCNA (a proliferative cell marker) (*Figure 2A* and *Figure 2—figure supplement 1A*). The result showed that as early as 24 hpi, the number of *cxcl18b*$^+$ MG was rapidly peaked with no emergence of proliferative MG (11±4, n=10 in *cxcl18b*$^+$ MG; mean ± SEM), and then *cxcl18b*$^+$ MG continued declining in number over time and reached the lowest level since 96 hpi (1±1, n=7; mean ± SEM; *Figure 2B*). In contrast, the number of proliferative MG (PCNA$^+$) peaked at 72 hpi and decreased to the lowest level at 120 hpi (9±2, n=11 in 72 hpi retina; 1±1, n=5 in 120 hpi retina; mean ± SEM; *Figure 2B*). Note that *cxcl18b*$^+$ MG was mostly proliferative at 72 hpi (*Figure 2—figure supplement 1B*).

To further verify the temporal expression of *cxcl18b* in MG following the cone ablation, we created a new transgenic reporter *Tg(cxcl18b: GFP)* by cloning a 3k-bp-long cis-element of 5'UTR with GFP, allowing real-time monitoring of injury-induced *cxcl18b* expression in vivo (*Figure 2C* and *Figure 2—figure supplement 1C and D*). Combining this line with PCNA immunostaining, we confirmed the remarkable increase of *cxcl18b* expression (GFP$^+$) at 48 hpi (0±1, n=4 in uninjured retinas vs 11±4, n=7 in 48 hpi; mean ± SEM; *Figure 2D and E*). Due to the prolonged stay of GFP protein, we could also observe that some GFP$^+$ MG were also PCNA$^+$ (*Figure 2D*). The result of proliferative MG as a subpopulation of *cxcl18b*$^+$ MG led to an outstanding question as to whether *cxcl18b*-defined MG transitional states represented an essential route to injury-induced proliferation.

To address it, we created a new transgene *Tg(cxcl18b: Cre-vmhc: mCherry:: ef1α: loxP-DsRed-loxP-EGFP:: lws2: nfsb-mCherry)* to perform the clonal analysis of historical *cxcl18b*-expressing in MG after the G/R cone ablation (details in Materials and methods; *Figure 2—figure supplement 1E*). After ablating the G/R cone by MTZ treatment at 6 dpf for 3 consecutive days, we found that the number of *cxcl18b* lineage-traced MG (marked by GFP$^+$) and PCNA$^+$ MG was significantly increased at 72 hpi (*Figure 2F*). Further analysis showed that all PCNA$^+$ MG were *cxcl18b* lineage-traced MG, indicating the *cxcl18b* lineage-traced MG were the ones who could eventually enter into the cell cycle (13±2 cells for PCNA$^+$ vs 13±2 cells PCNA$^+$ and GFP$^+$ MG; percentage of PCNA$^+$ and *cxcl18b*$^+$ vs PCNA$^+$=93 ± 2%; n=14; p=0.19; mean ± SEM; *Figure 2G*). Further analysis showed that at 120 hpi, the time point that the MG proliferation has largely ceased, GFP$^+$ MG with the lineage history of injury-induced *cxcl18b* expression constituted about 44% of GS$^+$ MG at the central retina, indicating that only about half of MG could enter *cxcl18b*$^+$ transitional states following the cone ablation (14±3 in GS$^+$ and GFP$^+$ MG; 28±4 in GS$^+$ and GFP$^-$ MG, n=14; p<0.05; mean ± SEM; *Figure 2I*). Together, our clonal analysis demonstrated that proliferative MG mostly originated from *cxcl18b*-defined MG transitional states, and 44% central MG could become *cxcl18b* positive. To investigate whether *cxcl18b*

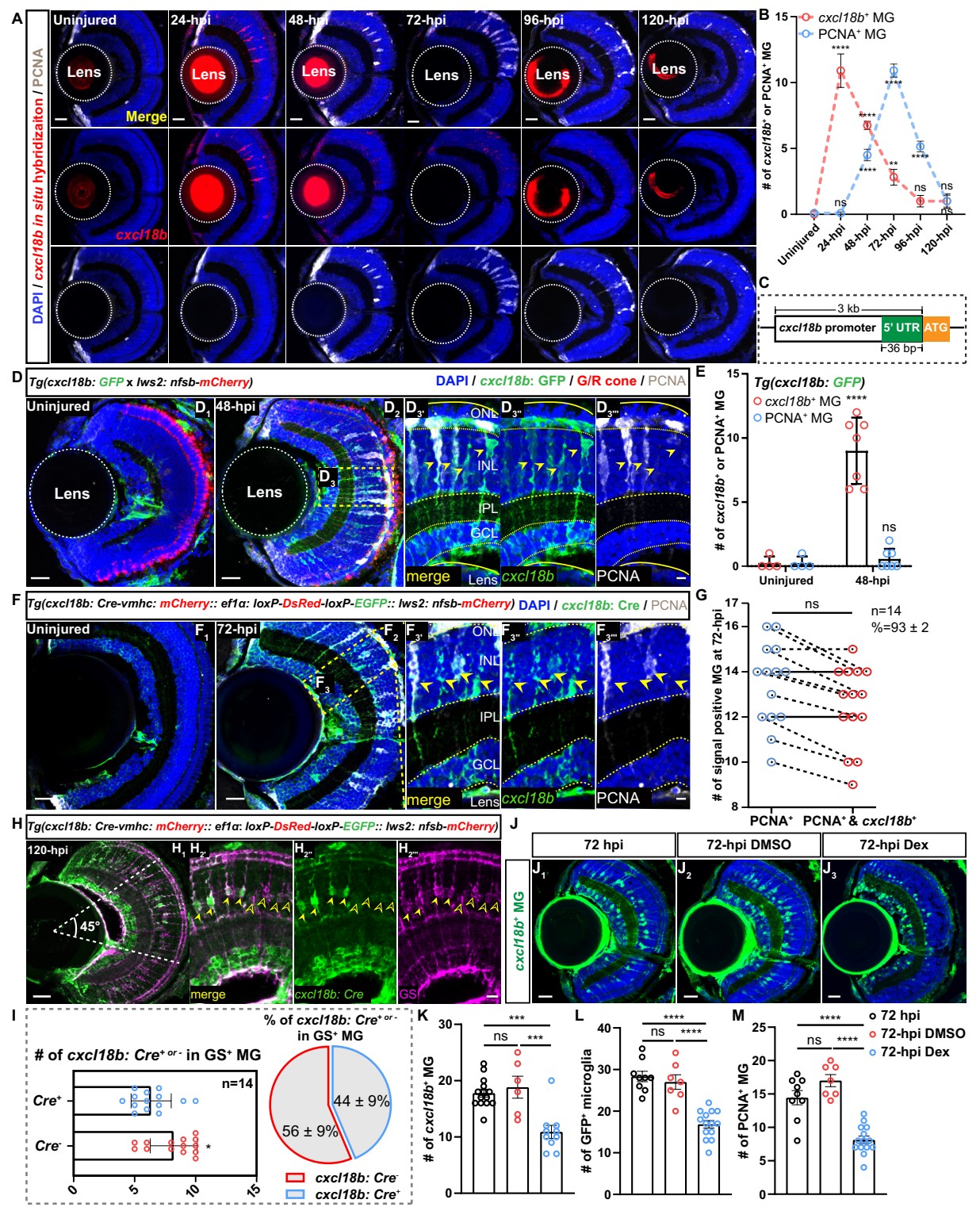

**Figure 2.** Clonal analysis reveals the proliferative Müller glia (MG) mostly originated from *cxcl18b*+ MG transitional states. (**A**) Representative images show dynamic expression of *cxcl18b* (red, in situ hybridization) and PCNA (white, immunostaining) in *Tg(lws2: nfsb-mCherry)* retina at different time points following the green/red (G/R) cone ablation. Scale bars: 20 μm. (**B**) Quantitative plots showing the number of *cxcl18b*+ (red curve, significance shown above the curve) and PCNA+ MG (blue curve, significance shown below the curve) in uninjured (n=11) and injured retina at 24 hr post-injury (hpi) (n=10), 48 hpi (n=12), 72 hpi (n=11), 96 hpi (n=7), and 120 hpi (n=5). Each injured time point was compared to the uninjured retina (mean ±

*Figure 2 continued on next page*

*Figure 2 continued*

SEM; ****p<0.0001, **p<0.01, ns, p>0.05; one-way ANOVA followed by Tukey's HSD test). (**C**) Schematic diagram of the *cxcl18b* promoter was used to construct the reporter fish line *Tg(cxcl18b: GFP)* and clonal analysis fish line *Tg(cxcl18b: Cre-vmhc-mCherry:: ef1α: loxP-DsRed-loxP-EGFP; lws2: nfsb-mCherry)*. (**D**) Immunostaining of injury-induced *cxcl18b*+ (green, indicated by *Tg(cxcl18b: GFP)*) and proliferative (PCNA+, white) MG showing overlapping in the central retina area (yellow arrows, GFP+/PCNA+ MG) at 48 hpi. The high-magnification images of the boxed area (**D₃– D₃'''**). The area of the retina is labeled with a dashed line, and each layer structure is labeled with dashed lines and marked with the outer nuclear layer (ONL), inner nuclear layer (INL), inner plexiform layer (IPL), ganglion cell layer (GCL), and lens. Scale bars: 20 μm (**D₁– D₂**) and 5 μm (**D₃– D₃'''**). (**E**) Quantitative plots showing the number of *cxcl18b*+ MG (red) and proliferative MG (PCNA+, blue) in (**D₁**) uninjured (n=4) and (**D₂**) 48 hpi retina (n=7) (mean ± SEM; ****p<0.0001, ns, p>0.05; two-way ANOVA followed by Tukey's HSD test). (**F**) Clonal analysis of injury-induced *cxcl18b*+ MG in transgenic fish line *Tg(cxcl18b: Cre-vmhc: mCherry:: ef1α: loxP-DsRed-loxP-EGFP:: lws2: nfsb-mCherry)* at 72 hpi retina showing overlapping between proliferative (PCNA+, white) MG with *cxcl18b*+ (green, yellow arrows). The high-magnification images of the boxed area (**F₃– F₃'''**). Scale bars: 20 μm (**F₁– F₂**) and 5 μm (**F₃– F₃'''**). (**G**) Quantitative analysis at 72 hpi shows no significance in the number of proliferative MG (PCNA+, blue) and double-positive (PCNA+/*cxcl18b*+, red) MG (n=14; mean ± SEM; ns, p>0.05; unpaired t-test) in (**F**), with 93±2% of PCNA+ MG also being *cxcl18b*+. (**H**) Representative images show that not all mature MG stained with glutamate synthase (GS+, magenta) are *cxcl18b*+ (green, labeled by *cxcl18b*: Cre) in the central retinal area (white dashed lines identified a 45° angular region originating from the optic nerve). The high-magnification images of the boxed area (**H₂– H₂'''**). Scale bars: 20 μm (**H₁**) and 5 μm (**H₂– H₂'''**). (**I**) Quantification of GS+/*cxcl18b*: Cre+ double-positive (blue, yellow arrows in **H**) and GS+/*cxcl18b*: Cre⁻ single-positive (red, open yellow arrowheads in **H**) MG (n=14, mean ± SEM; *p<0.05; one-way ANOVA followed by Tukey's HSD test), and the proportion of *cxcl18b*: Cre⁺ ᵒʳ ⁻ MG within the total population of mature (GS+) MG in the central retina. (**J₁– J₃**) Representative images showing injury-induced *cxcl18b*+ MG (green) in *Tg(lws2: nfsb-mCherry)* cross with *Tg(cxcl18b: GFP)* fish retina treated with dexamethasone (Dex) or DMSO at 72 hpi. Scale bars: 20 μm. (**K – M**) Quantitative plots showing the number of *cxcl18b*+ MG (72 hpi: n=14; DMSO: n=6, Dex: n=10) in J₁-J₃; recruited microglia in ***Figure 2—figure supplement 1J1–J3*** and proliferative MG in ***Figure 2—figure supplement 1J4–J6*** (72 hpi: n=9; DMSO: n=7, Dex: n=14) in *Tg(mpeg1: GFP; lws2: nfsb-mCherry)* retinas after DMSO or Dex treatment at 72 hpi (mean ± SEM; ****p<0.0001, ***p<0.001, ns, p>0.05; one-way ANOVA followed by Tukey's HSD test).

The online version of this article includes the following source data and figure supplement(s) for figure 2:

**Source data 1.** Quantitative analysis of *cxcl18b* in situ hybridization signal and PCNA+ Müller glia (MG) in uninjured and injured retinas at the indicated time points.

**Source data 2.** Quantification of the number of *cxcl18b*+ and PCNA+ Müller glia (MG) in the uninjured and 48 hr post-injury (hpi) zebrafish retinas from *Tg(cxcl18b: GFP)* fish.

**Source data 3.** Quantification of the number of *cxcl18b*+ and PCNA+/*cxcl18b*+ double-positive Müller glia (MG) in injured zebrafish retinas in the lineage-tracing experiment.

**Source data 4.** Quantification of GS+/*cxcl18b*: Cre⁺ and GS+/*cxcl18b*: Cre⁻ Müller glia (MG) in the central retina region.

**Source data 5.** Quantification of *cxcl18*+ Müller glia (MG), microglia, and PCNA+ MG in the immunosuppression experiment.

**Figure supplement 1.** Gene description of *cxcl18b* does not reduce Müller glia (MG) proliferation.

**Figure supplement 1—source data 1.** Quantification of PCNA+ Müller glia (MG) in the injured retinas under control and *cxcl18b* disruption conditions.

was required for MG proliferation following G/R cone ablation, we employed CRISPR-Cas9-mediated gene disruption, using two sgRNAs targeting *cxcl18b* (***Figure 2—figure supplement 1F and G***). We found that *cxcl18b* knockout did not reduce MG proliferation after G/R cone ablation at 72 hpi (13±3, n=11 in WT; 11±3, n=7 in *scramble* sgRNA-injected; 13±3, n=7 in *cxcl18b* sgRNA-injected; mean ± SEM), suggesting that *cxcl18b* per se does not regulate MG proliferation directly (***Figure 2—figure supplement 1H and I***). This led us to wonder about the induction of *cxcl18b*-defined MG transitional states.

As an inflammatory chemokine, *cxcl18b* serves as a reliable marker of inflammation and regulates neutrophil recruitment to injury sites (***Goumenaki et al., 2024***; ***Torraca et al., 2017***). Inflammation has been previously shown to be critical for inducing regenerative responses in adult zebrafish, where it promotes reactive microglia/macrophages and MG proliferation in the retina (***Iribarne and Hyde, 2022***; ***Kyritsis et al., 2012***). Notably, suppressing the immune response using dexamethasone (Dex) in zebrafish retina reduced microglial reactivation and significantly decreased the number of proliferative MG (***Silva et al., 2020***; ***Zhang et al., 2020***). In our study, we identified the *cxcl18b*-defined transitional states as the essential routing for MG proliferation after G/R cone ablation. These results prompted us to investigate whether the inflammatory responses mediated by recruited microglia are indispensable for the formation of these *cxcl18b*-defined transitional states.

To address this, we examined *cxcl18b* expression using *Tg(cxcl18b: GFP)* after inhibiting inflammation using Dex (***Iribarne and Hyde, 2022***) and observed a significant reduction in the number of *cxcl18b*+ MG (GFP+ cells) at 72 hpi (11±4, n=10 in Dex-treated retina vs 19±5, n=6 in DMSO treatment; and 18±3, n=14 in the 72 hpi retina; mean ± SEM) (***Figure 2K***). Consistent with earlier

reports, we observed that Dex treatment inhibited the migration of microglia (indicated by *Tg(mpeg1: GFP)*); 16±4, n=14 in Dex-treated retina vs 27±5, n=7 in DMSO treatment; and 28±4, n=9 in the 72 hpi retina; mean ± SEM) to the ONL and significantly reduced the number of proliferative MG (PCNA+; 8±2, n=14 in Dex-treated retina vs 17±2, n=7 in DMSO treatment; and 14±3, n=9 in the 72 hpi retina; mean ± SEM) at 72 hpi after G/R cone ablation (*Figure 2L and M* and *Figure 2—figure supplement 1J*). These findings suggest that microglia-mediated inflammation may contribute to the activation of *cxcl18b*-defined transitional states that precede MG proliferation, although a causal relationship remains to be established. While Dex suppressed both microglial recruitment and *cxcl18b*+ MG generation, its broad anti-inflammatory action precludes definitive conclusions about microglial causality. Dissecting this relationship would require concurrent ablation of microglia and cone photoreceptors using a triple-transgenic strategy, which is beyond the scope of the current study. Targeted approaches will be necessary to resolve the specific role of microglia in initiating *cxcl18b* expression.

### *cxcl18b*-defined MG transitional states recapitulate molecular features of retinal stem cells in the CMZ

Interestingly, we observed the *cxcl18b* expression in the CMZ after the cone ablation besides its high expression in the MG (*Figure 2A*). We were then curious about the *cxcl18b* expression in the developmental retina, as well as in the CMZ without the injury. Notably, in situ results showed that *cxcl18b* was largely absent from the central region but presented in the peripheral region of 30 hpf retinae, whereas it was highly expressed in the most peripheral region of the CMZ, where *fabp11a* and *col15α1b*, two putative markers for postembryonic retinal stem cells (RSCs), are located (*Gonzalez-Nunez et al., 2010*; *Raymond et al., 2006*; *Figure 3A-B and E-F*, *Figure 3—figure supplement 1A*). The transgenic line of *Tg(cxcl18b: GFP)* also showed a robust *cxcl18b* expression in the CMZ (*Figure 3—figure supplement 1D*). Consistently, our scRNA-seq data of CMZ cells also confirmed the co-expression of *cxcl18b*, *fabp11a*, and *col15α1b* (*Figure 3C–D and G–H*, *Figure 3—figure supplement 1B*). Furthermore, cluster 1 MG, at the earlier stage of transitional states, has the highest *cxcl18b* expression with *col15α1b* expression (*Figure 1F*, *Figure 3—figure supplement 1C*). All these results suggested that *cxcl18b*-defined transitional states, at least to some extent, represent the developmental state of retinal stem cells in the CMZ, but not that of embryonic retinal progenitors.

### *Nos2b* is required for MG entry into the proliferation via NO signaling

Previous studies have demonstrated the involvement of redox signaling in cell regeneration processes in various tissues across species (*Han et al., 2014*; *Hunter et al., 2018*; *Matrone et al., 2021*; *Yoo et al., 2012*). All this evidence led us to directly test the roles of redox genes in serving as the molecular mechanism underlying injury-induced MG proliferation. Thus, we first examined the expression levels of a comprehensive list of redox genes in *cxcl18b*-defined MG transitional states (Cluster 1/2/5) in our scRNA-seq data (*Figure 1H*) and screened the influence of 18 genes from each major category of redox signaling on injury-induced MG proliferation using CRISPR-Cas9-mediated gene disruption (*Figure 4—figure supplement 1A*). We focused on the NO signaling pathway, targeting three genes encoding NO synthases (*Nos*): neuronal *Nos* (*nos1*) and two inducible forms (*nos2a* and *nos2b*), as well as the gene encoding S-nitrosoglutathione reductase (*gsnor* or *adh5*), which modulates reactive NO signaling (*Figure 4A* and *Figure 4—figure supplement 1B*). The consequence and efficiency of gene disruption were verified by DNA sequencing (*Figure 4—figure supplement 1C*). Notably, the disruption of *nos2b* resulted in a significant reduction of PCNA+ MG at 72 hpi (6±2, n=22 in *nos2b*-disrupted vs 11±3, n=7 in *scramble* sgRNA-injected; mean ± SEM) (*Figure 4B*). Noted that *nos* gene disruption did not significantly alter microglia recruitment or G/R cone ablation at 72 hpi, suggesting that the influence of NO on injury-induced MG proliferation was not via inflammatory reactions of recruited microglia or injury degree (*Figure 4—figure supplement 1D–F*).

To further examine the function of *Nos2b* via NO, we employed various *Nos* inhibitors (L-NG-nitro arginine methyl ester, L_NAME; L-NG-monomethyl arginine, L_NMMA; 1400W) and NO scavengers (carboxy-PTIO, C-PTIO) (*Goldstein et al., 2003*; *Hong et al., 2012*; *Moore et al., 1990*; *Rees et al., 1990*). We performed the intraocular injection of the drugs (PBS as control) into the zebrafish eye from 2 days before cell ablation until 72 hpi (*Figure 4C*). Notably, the NO scavenger C-PTIO mostly suppressed MG proliferation, indicating the involvement of NO (cell number of PCNA+ MG: 3±2, n=27 after blocking NO by C-PTIO vs 14±4, n=16 in PBS-injected retina; mean ± SEM) (*Figure 4D and*

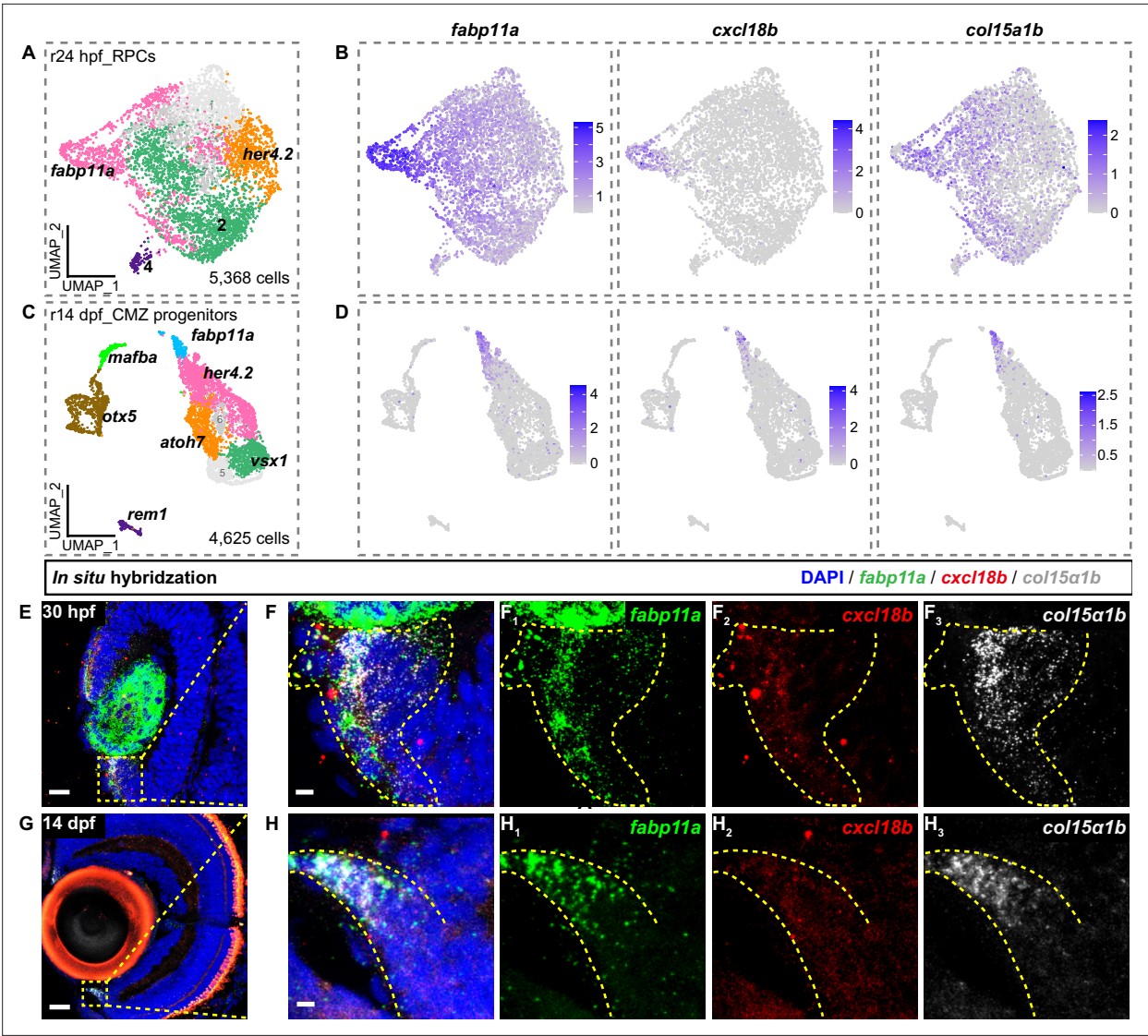

**Figure 3.** The *cxcl18b*-defined Müller glia (MG) transitional states recapitulate the developmental states of retinal stem cells (RSCs) in the ciliary marginal zone (CMZ). (**A, C**) UMAP plots display 5368 retinal progenitor cells (RPCs) at 24 hr post-fertilization (hpf) and 4625 CMZ progenitor cells at 14 days post-fertilization (dpf). Clusters are indicated by their cluster-specific marker genes based on previously published single-cell RNA-sequencing (scRNA-seq) data (*Xu et al., 2020*). (**B, D**) UMAP plots showing expression of *fabp11a*, *cxcl18b*, and *col15α1b* at r24-hpf RPCs (embryonic states) (**B**) and r14-dpf CMZ-progenitors (postembryonic states) (**D**). (**E – H₃**) In situ hybridization images showing the expression of *fabp11a* (green), *col15α1b* (white), two putative markers for postembryonic RSCs, and *cxcl18b* at 30 hpf (**E – F₃**) and 14 dpf (**G – H₃**) retina. The high-magnification images of the boxed area (**F – F₃, H – H₃**). The area of these three in situ signal trouble positives is labeled with a dashed yellow line. Scale bars: 20 µm (**E, G**) and 3 µm (**F – F₃, H – H₃**).

The online version of this article includes the following figure supplement(s) for figure 3:

**Figure supplement 1.** UMAP plots show the expression of cluster-specific marker genes utilized to identify developmental states.

*E*). Moreover, 1400W (an inhibitor specific to inducible *Nos*, 7±2, n=10; mean ± SEM) and L_NAME (a broad inhibitor to all three *Nos* forms, 8±2, n=12; mean ± SEM) could also significantly reduce the number of proliferative MG after the ablation, whereas L_NMMA (the inhibitor specific to neuronal *Nos*) did not influence MG proliferation (11±3, n=12; mean ± SEM) (*Figure 4E*). Taken together, these results highlight the critical role of NO signaling in regulating injury-induced MG proliferation.

We further generated NO pathway mutant zebrafish (*nos1*, *nos2a*, *nos2b*, and *gsnor*) to investigate the role of NO in MG proliferation following G/R cone ablation. Utilizing CRISPR/Cas9-mediated gene disruption, we successfully screened out *nos1*, *nos2a*, *nos2b*, and *gsnor* mutants, characterized by deletions of 133 bp, 13 bp, 220 bp, and 11 bp coding sequences, respectively (see Materials and

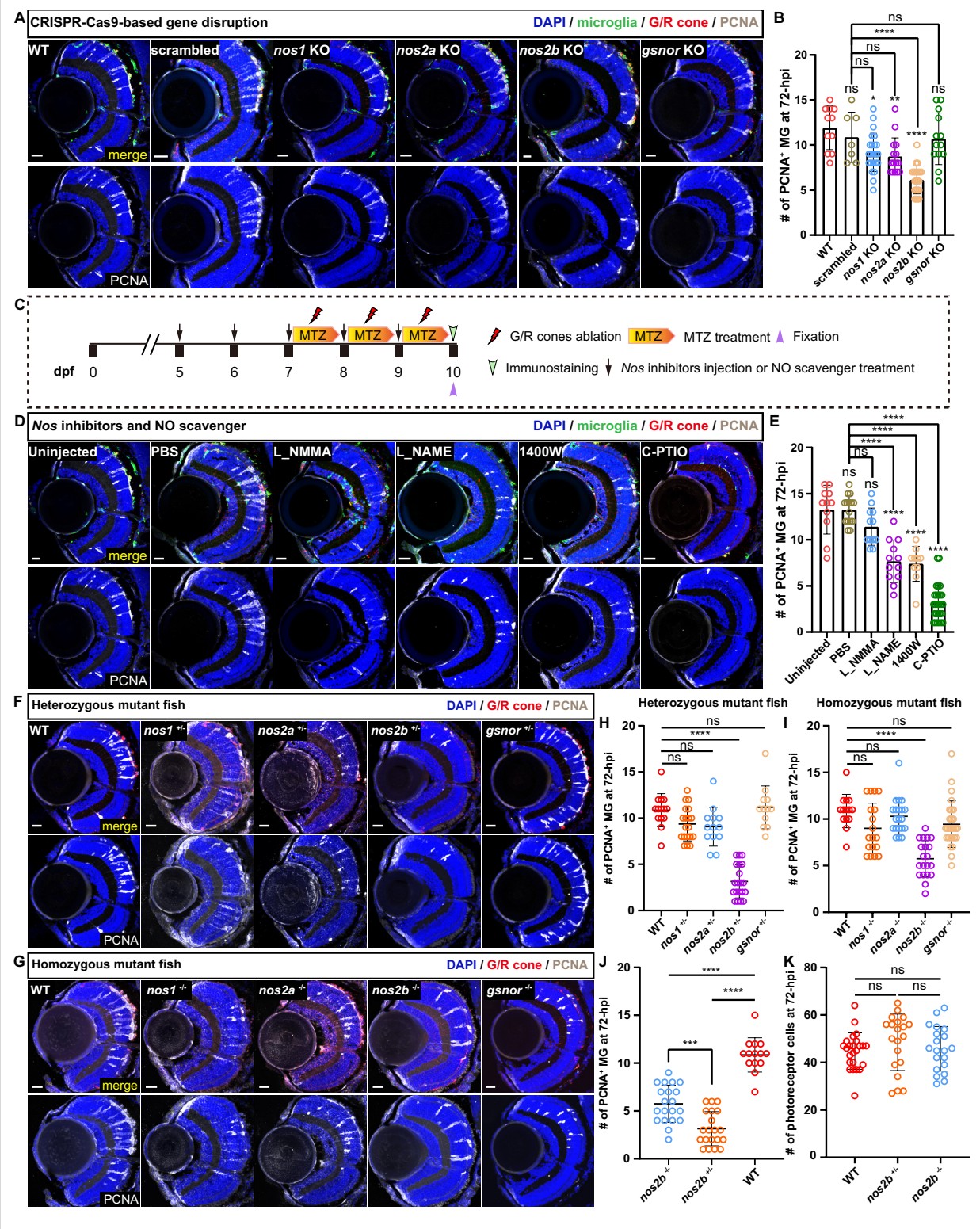

**Figure 4.** The nitric oxide metabolic pathway regulates injury-induced Müller glia (MG) proliferation. (**A**) Representative images of microglia recruitment (green, marked by *Tg(mpeg1: GFP)*), green/red (G/R) cone ablation (red, marked by *Tg(lws2: nfsb-mCherry)*) and proliferative MG (white, marked by PCNA$^+$) at 72 hr post-injury (hpi) with nitric oxide metabolism pathway genes disruption (*nos1/nos2a/nos2b/gsnor*). Scale bars: 20 μm. (**B**) Quantitative analysis of the number of proliferative MG (PCNA$^+$) at 72 hpi in (**A**). In total, we collected 11 retinas for wild-type (WT) (n=11), *scramble* sgRNA-injection (n=7), *nos1* sgRNA-injection (n=23), *nos2a* sgRNA-injection (n=15), *nos2b* sgRNA-injection (n=22), and *gsnor* sgRNA-injection (n=13) (mean ± SEM;

*Figure 4 continued on next page*

*Figure 4 continued*

****p<0.0001, **p<0.01, *p<0.05, ns, p>0.05; one-way ANOVA followed by Tukey's HSD test). (**C**) Schematic showing the experimental procedure of nitric oxide synthase (*Nos*) inhibitors injection or NO scavengers treatment in *Tg(lws2: nfsb-mCherry × mpeg1: GFP)* retina starting from 5 days post-fertilization (dpf) for 5 consecutive days to 10 dpf with 3 consecutive days of metronidazole (MTZ) treatment for G/R cone ablation from 7 dpf to 10 dpf. *Nos* inhibitors, NO scavengers, and MTZ solution were refreshed every 24 hr, and fish fixation was at 10 dpf for further immunostaining. (**D**) Representative images of microglial recruitment (green, marked by *Tg(mpeg1: GFP)*), G/R cone ablation (red, marked by *Tg(lws2: nfsb-mCherry)*), and proliferative MG (white, marked by PCNA$^+$) at 72 hpi following L-NMMA (10 mM), L-NAME (10 mM), 1400W (200 nM) intraocular injection, and PBS as control, or C-PTIO (10 mM) treatment. Scale bars: 20 μm. (**E**) Quantitative plots showing the number of proliferative (PCNA$^+$) MG at 72 hpi in (**D**). Retinas analyzed WT (n=11), PBS-injected (n=16), L_NMMA-injected (n=12), L_NAME-injected (n=12), 1400W-injected (n=10), and C-PTIO treatment (n=27) (mean ± SEM; ****p<0.0001, ns, p>0.05; one-way ANOVA followed by Tukey's HSD test). (**F–G**) Representative images of proliferative MG (PCNA$^+$, white) and G/R cone ablation (marked by *Tg(lws2: nfsb-mCherry)*, red) at 72 hpi in heterozygous (*nos1$^{+/-}$*, *nos2a$^{+/-}$*, *nos2b$^{+/-}$*, *gsnor$^{+/-}$*) (**F**) and homozygous mutants (**G**) of nitric oxide metabolism pathway genes (*nos1$^{-/-}$*, *nos2a$^{-/-}$*, *nos2b$^{-/-}$*, *gsnor$^{-/-}$*). Scale bars: 20 μm. (**H–I**) Quantitative plots showing the number of proliferative MG (white, PCNA$^+$) at 72 hpi in *nos* and *gsnor* mutant fish. In heterozygous (**H**), analyzed retinas include WT (n=14), *nos1$^{+/-}$* (n=19), *nos2a$^{+/-}$* (n=13), *nos2b$^{+/-}$* (n=20), *gsnor$^{+/-}$* (n=12). For homozygous (**I**), analyzed retinas include *nos1$^{-/-}$* (n=18), *nos2a$^{-/-}$* (n=20), *nos2b$^{-/-}$* (n=20), *gsnor$^{-/-}$* (n=27) (mean ± SEM; ****p<0.0001, ns, p>0.05; one-way ANOVA followed by Tukey's HSD test). (**J–K**) Quantitative plots showing the number of proliferative MG (white, PCNA$^+$) (**J**) and photoreceptor cells remain (**K**) at 72 hpi in *nos2b* hetero- or homozygous mutants (mean ± SEM; ****p<0.0001, ***p<0.001, ns, p>0.05; one-way ANOVA followed by Tukey's HSD test).

The online version of this article includes the following source data and figure supplement(s) for figure 4:

**Source data 1.** Quantitative analysis of PCNA$^+$ Müller glia (MG) in the *Nos* metabolism gene disruption experiment after retinal injury.

**Source data 2.** Quantitative analysis of PCNA$^+$ Müller glia (MG) in the *Nos* inhibitor or NO scavenger injection experiment after injury.

**Source data 3.** Quantification of the number of PCNA$^+$ Müller glia (MG) and photoreceptor cells in the retinas of fish with *Nos* mutations.

**Figure supplement 1.** Genetic disruption of the nitric oxide (NO) pathway modulates green/red (G/R) cone ablation and microglial recruitment.

**Figure supplement 2.** The genotype of nitric oxide metabolism pathway mutants.

methods for details, *Figure 4—figure supplement 2A and B*). Consistent with the gene disruption experiments described above, we observed a significant reduction in the number of proliferative (PCNA$^+$) MG at 72 hpi following G/R cone ablation in both heterozygous and homozygous *nos2b* mutants (*nos2b$^{+/-}$*: 3 ± 1, n = 20; *nos2b$^{-/-}$*: 6 ± 2, n = 20) compared to WT controls (11 ± 2, n = 14; mean ± SEM). In contrast, no significant changes in MG proliferation were observed in *nos1*, *nos2a*, or *gsnor* mutants compared to wild type (WT) (*Figure 4F–I*).

Interestingly, the reduction in proliferative MG was more pronounced in *nos2b* heterozygous mutants (*nos2b$^{+/-}$*) than in homozygous mutants (*nos2b$^{-/-}$*) (*nos2b$^{+/-}$* vs *nos2b$^{-/-}$*; p<0.001; mean ± SEM) (*Figure 4J*). We observed no significant difference in the loss of cone photoreceptor at 72 hpi between *nos2b* mutants and WT, indicating that the reduced MG proliferation observed in *nos2b* mutants is independent of the injury (WT: 45 ± 8 remaining cones, n = 24; *nos2b$^{+/-}$*: 49 ± 12, n = 20; *nos2b$^{-/-}$*: 46 ± 9, n = 20; mean ± SEM) (*Figure 4K*). This unexpected result suggests a concentration-dependent effect of NO on proliferative MG. Specifically, compared to homozygous mutants, heterozygous mutants with intermediate NO levels more effectively suppressed MG proliferation, whereas WT animals with higher NO levels promoted MG proliferation. This concentration-response pattern highlights the role of NO as a regulator, rather than a mediator, of injury-induced MG proliferation.

## Specific *nos2b* expression in *cxcl18b*-defined transitional states MG after G/R cone ablation

To further examine whether the expression of *nos2b* in *cxcl18b$^+$* MG, we used the newly generated transgene fish line in our study *Tg(cxcl18b: GFP)*, enabling us to monitor *cxcl18b$^+$* MG after the G/R cone ablation in a real-time manner. By crossing different fish lines, we fluorescently sorted out three post-injury MG populations (72 hpi MG, 72 hpi PCNA$^+$ MG, and 72 hpi *cxcl18b$^+$* MG) and three control groups (uninjured MG, 72 hpi retinal cells other than MG, and 72 hpi G/R cone) (*Figure 5A–C*). The real-time quantitative PCR (RT-qPCR) analysis showed that compared to the three control groups, the expression of *nos2b* was significantly higher in *cxcl18b$^+$* MG than in 72 hpi MG and 72 hpi PCNA$^+$ MG (relative expression of *nos2b*: 89±32, repeats n=7 in *cxcl18b$^+$* MG; 14±17, repeats n=5 in 72 hpi MG; and 1±1, repeats n=6 in PCNA$^+$ MG; mean ± SEM) (*Figure 5D* and *Figure 5—figure supplement 1D–F*). Together with the fact that *cxcl18b$^+$* MG contained PCNA$^+$ and PCNA$^-$ populations in *Tg(cxcl18b: GFP)* (*Figure 2D*), our result indicated that *nos2b* was specifically expressed in *cxcl18b$^+$* non-proliferative MG, which agreed with the scRNA-seq result of the specific expression of *cxcl18b* in

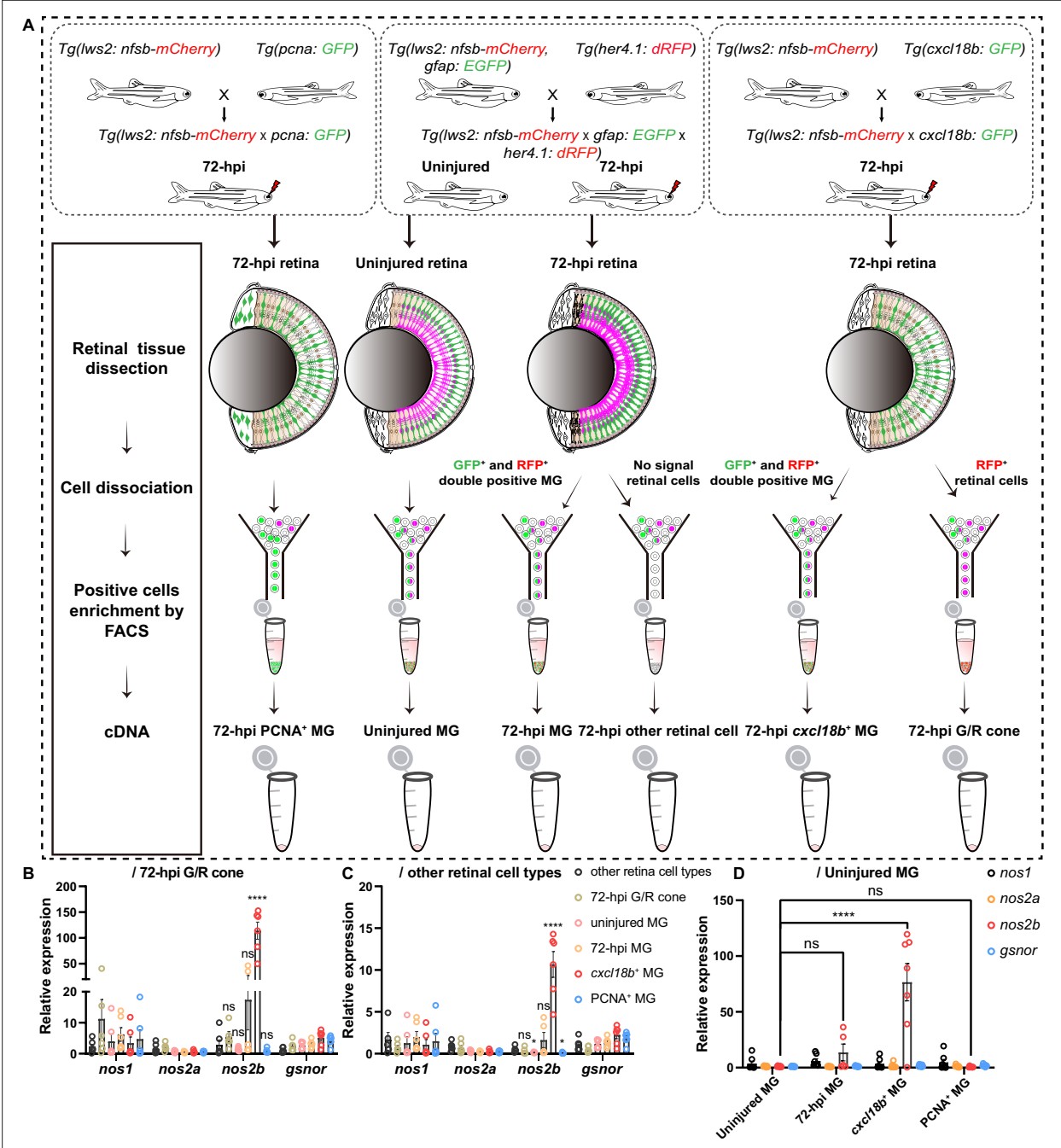

**Figure 5.** Real-time quantitative PCR (RT-qPCR) analysis reveals *nos2b* cell-specific expression in the injury-induced *cxcl18b*+ Müller glia (MG). (**A**) Schematic showing the workflow for isolating and enriching for three post-injury MG populations (72 hr post-injury [hpi] MG, 72 hpi PCNA+ MG, and 72 hpi *cxcl18b*+ MG) and three control groups (uninjured MG, 72 hpi retinal cells other than MG, and 72 hpi green/red [G/R] cones) using fluorescence-activated cell sorting (FACS). (**B–C**) RT-qPCR analysis of *nos1*, *nos2a*, *nos2b*, and *gsnor* expression in different cell populations. Expression levels are shown relative to 72 hpi G/R cones (**B**) and other 72 hpi retinal cell types (**C**). A total of six independent replicates were performed for cell population enrichment and cDNA template preparation (n=6, mean ± SEM; ****p<0.0001, ns, p>0.05; two-way ANOVA followed by Tukey's HSD test). (**D**) RT-qPCR analysis comparing *nos1*, *nos2a*, *nos2b*, and *gsnor* expression in distinct MG states relative to uninjured MG (repeats n=7 in *cxcl18b*+ MG; n=5 in 72 hpi MG; and repeats n=6 in PCNA+ MG; n=6 in the uninjured retina; mean ± SEM; ****p<0.0001, ns, p>0.05; two-way ANOVA followed by Tukey's HSD test).

The online version of this article includes the following source data and figure supplement(s) for figure 5:

**Source data 1.** Real-time quantitative PCR (RT-qPCR) analysis of *nos1*, *nos2a*, *nos2b*, and *gsnor* expression across retinal cell populations and distinct Müller glia (MG) states.

**Figure supplement 1.** In situ hybridization reveals *nos2b* cell-specific expression in the *cxcl18b*-defined transitional Müller glia (MG) states.

*Figure 5 continued on next page*

*Figure 5 continued*

**Figure supplement 1—source data 1.** Real-time quantitative PCR (RT-qPCR) analysis of *glula*, *glulb*, *cxcl18b*, and *pcna* expression across retinal cell populations and Müller glia (MG) states.

three transitional states (Clusters 1, 2, and 5; *Figure 1H*). In situ hybridization using an HCR molecular probe in the *Tg(lws2: nfsb-mCherry:: cxcl18b: GFP)* fish line also suggests that injury-induced *nos2b* expression was specific in the *cxcl18b*-defined transitional state MG (*Figure 5—figure supplement 1A–C3*). Thus, our analysis reveals that *nos2b* was specifically expressed in *cxcl18b*⁺ transitional MG states, particularly in non-proliferative cells.

## NO produced by *nos2b* determines *cxcl18b*⁺ MG entry for proliferation

To further explore the role of *cxcl18b*⁺ MG-specific *nos2b* in regulating the entry of MG proliferation after the G/R cone ablation, we developed a sophisticated clonal analysis of *cxcl18b*⁺ MG with genetic disruption of *nos2b* using CRISPR-Cas9 method. To achieve glial type-specific gene manipulation, we employed an adenovirus strain that specifically infects radial glia in zebrafish (*Jia et al., 2019*; *Liu et al., 2021*) and confirmed that it could faithfully mark MG in the zebrafish retina (*Figure 6A and B*). For the clonal analysis, we performed the intraocular injection of two viruses packaged with elements of *cxcl18b: gal4* and *UAS: Cas9-T2A-Cre-u6: sgRNA (nos2b)* into the eye of *Tg(lws2: nfsb-mCherry:: ef1α: loxP-DsRed-loxP-EGFP)* at 5 dpf, virus packaged with the element of *UAS: Cas9-T2A-Cre-u6: empty* as the control (*Figure 6C and D* and *Figure 6—figure supplement 1A*). One day after the virus infection, the fish was treated with MTZ for 3 consecutive days to ablate the G/R cone (*Figure 6E*).

At 72 hpi, we collected the clones (GFP⁺) derived from *cxcl18b* lineage-traced MG and analyzed their proliferative property (*Figure 6F and G*). The analysis of virus-infected clones (GFP⁺) cell-cycle re-entry (PCNA⁺) after G/R cone ablation revealed that only ~23% of GFP⁺ MG clones remained PCNA⁺ in the *nos2b* knockout group, compared to ~75% in the control group (*Figure 6H*). Note that the efficiency of *nos2b* sgRNAs was confirmed in terms of mutation types and knockout efficiency by sequencing (*Figure 6—figure supplement 1B–D*).

Taken together, these findings suggested that it is the NO produced by *Nos2b* within *cxcl18b*-defined transitional state MG that specifically drives MG from the transitional state into proliferation following injury, highlighting a pivotal mechanism underlying injury-induced regenerative processes.

## NO decreased Notch activity that is responsible for injury-induced MG proliferation

Previous studies have shown that *ascl1α* and Notch signaling are essential for MG proliferation in the injured zebrafish retina (*Conner et al., 2014*; *Wan et al., 2012*). Regarding Notch signaling, a high Notch3 expression is reported to maintain MG quiescence. In response to the injury, *notch3* expression is downregulated, but *notch1a* is necessary for the continued proliferation of the progenitors (*Campbell et al., 2022*). Consistently, we observed that *notch3* and *hey* (the Notch downstream target) were highly expressed in uninjured MG clusters and became reduced from the early transitional state (Cluster 4) to the proliferative states (Clusters 3 and 6), whereas *notch1a/1b* and *ascl1α* were prominently expressed in the late stage of *cxcl18b*⁺ transitional states (Cluster 5) and the proliferative states (Cluster 3; *Figures 1 and 7*). Interestingly, upstream regulators of Notch signaling activation, such as *fgf8a*, *fgf8b* (*Wan and Goldman, 2017*), and *tgfb3* (*Lee et al., 2020*), were predominantly expressed in Clusters 4 and 1, preceding the expression of *cxcl18b* (*Figure 7A*). These results led us to wonder whether NO regulated *cxcl18b*-defined transitional state MG cell-cycle re-entry via the Notch signaling pathway.

To examine the influence of NO signaling blockage on Notch activity dynamics following the cone ablation, we employed a reporter line *Tg(Tp1bglob: EGFP)* (referred to as *Tg(Tp1:EGFP)*), in which EGFP is driven by the TP1 element, the direct target of the intracellular domain of Notch receptors (NICD) that is generated upon Notch activation (*Parsons et al., 2009*; *Quillien et al., 2014*). We treated fish with the NO scavenger C-PTIO and MTZ, starting at 5 dpf for 5 continuous days and at 7 dpf for 3 continuous days, respectively, followed by immunostaining for EGFP and PCNA (*Figure 7B*). Interestingly, *Tp1*: EGFP⁺ MG were significantly reduced at all three injury time points (cell number of EGFP⁺ clones: 34±3, n=9 in 24 hpi retina; 29±3, n=4 in 48 hpi retina; 33±4, n=6 in 72 hpi retina vs 42±1, n=4 in the uninjured retina; mean ± SEM), demonstrating a decrease in Notch activity following

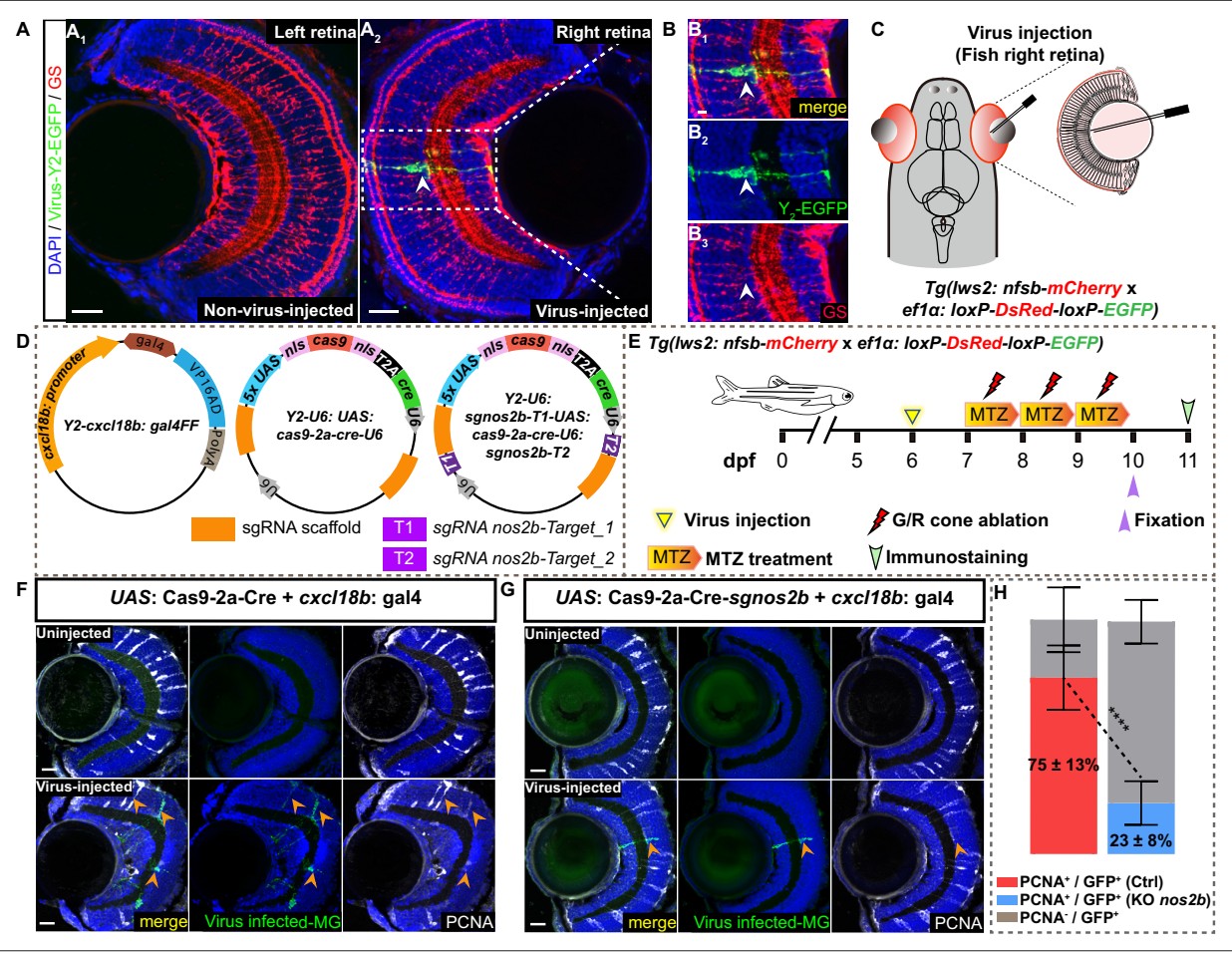

**Figure 6.** Nitric oxide (NO) produced by *nos2b* in *cxcl18b*⁺ Müller glia (MG) regulates injury-induced proliferation. (**A–C**) Representative images showing the adenovirus-mediated infection (green, indicated by Y₂-GFP) specifically target MG (red, GS staining) in the zebrafish retina. The virus was intraocularly injected into the right eye of *Tg(lws2: nfsb-mCherry x ef1α: loxP-DsRed-loxP-EGFP)* fish (**A₂, C**), with the left eye as a wild-type (WT) control (**A₁**). The high-magnification images of the boxed area (B–B₃). Scale bars: 20 μm (**A₁, A₂**) and 5 μm (**B–B₃**). (**D**) Schematic showing the design of the *cxcl18b*⁺ MG-specific *nos2b* knockout system. The viral construct consists of three plasmids: (1) *gal4* expression driven by the *cxcl18b* promoter; (2) *UAS*-derived *Cas9* and *Cre* elements, and (3) *U6* promoters driving two sgRNAs targeting *nos2b*, with a non-targeting sgRNA as the control. (**E**) Schematic showing the procedure of injury process and intraocular viral injection in *Tg(lws2: nfsb-mCherry × ef1α: loxP-DsRed-loxP-EGFP)* fish. (**F–G**) Representative images showing proliferative MG (PCNA⁺, white) with *cxcl18b*⁺ MG-specific knockout *nos2b* in (**G**) and control in (**F**), (GFP⁺, green, yellow arrows) are defined as virus-infected clones. Upper panels show WT retina (no virus injected). Bottom panels show retinas injected with the virus (two sgRNA targets as *nos2b* knockout and without sgRNA targets as control). Scale bars: 20 μm. (**H**) Quantification of proliferative (PCNA⁺/GFP⁺, red bars) and non-proliferative (PCNA⁻/GFP⁺, gray bars) MG clones in (**F**). For control, ~75% of virus-infected clones entered the cell cycle (PCNA⁺; red bars), with 90/120 clones analyzed across 8 independent experiments (n=8). For *nos2b* knockout clones, ~23% entered the cell cycle (PCNA⁺; blue bars), with 22/103 clones analyzed across 6 independent experiments (n=6) (mean ± SEM; ****p<0.0001; two-way ANOVA followed by Tukey's HSD test).

The online version of this article includes the following source data and figure supplement(s) for figure 6:

**Source data 1.** Quantification of proliferative (PCNA⁺/GFP⁺) and non-proliferative (PCNA⁻/GFP⁺) Müller glia (MG) clones in control and *nos2b* MG-specific knockout conditions.

**Figure supplement 1.** Analysis of cell-specific knockout efficiency for *nos2b* in virus-infected clones.

**Figure supplement 1—source data 1.** PDF file containing original DNA gel for *Figure 6—figure supplement 1C*, indicating the relevant bands.

**Figure supplement 1—source data 2.** Original files for DNA gel analysis displayed in *Figure 6—figure supplement 1C*.

G/R cone ablation (*Figure 7C*). Notably, this reduction in Notch activation was further rescued by NO blocking using C-PTIO (cell number of EGFP⁺ clones: 42±1, n=6 in C-PTIO-treated retina at 24 hpi; 42±2, n=5 at 48 hpi; 44±2, n=7 at 72 hpi; mean ± SEM), suggesting that NO modulates Notch signaling (*Figure 7C*). Meanwhile, C-PTIO treatment significantly reduced the number of proliferative MG (marked by PCNA) (*Figure 7D*). These findings indicated that injury-induced NO suppresses

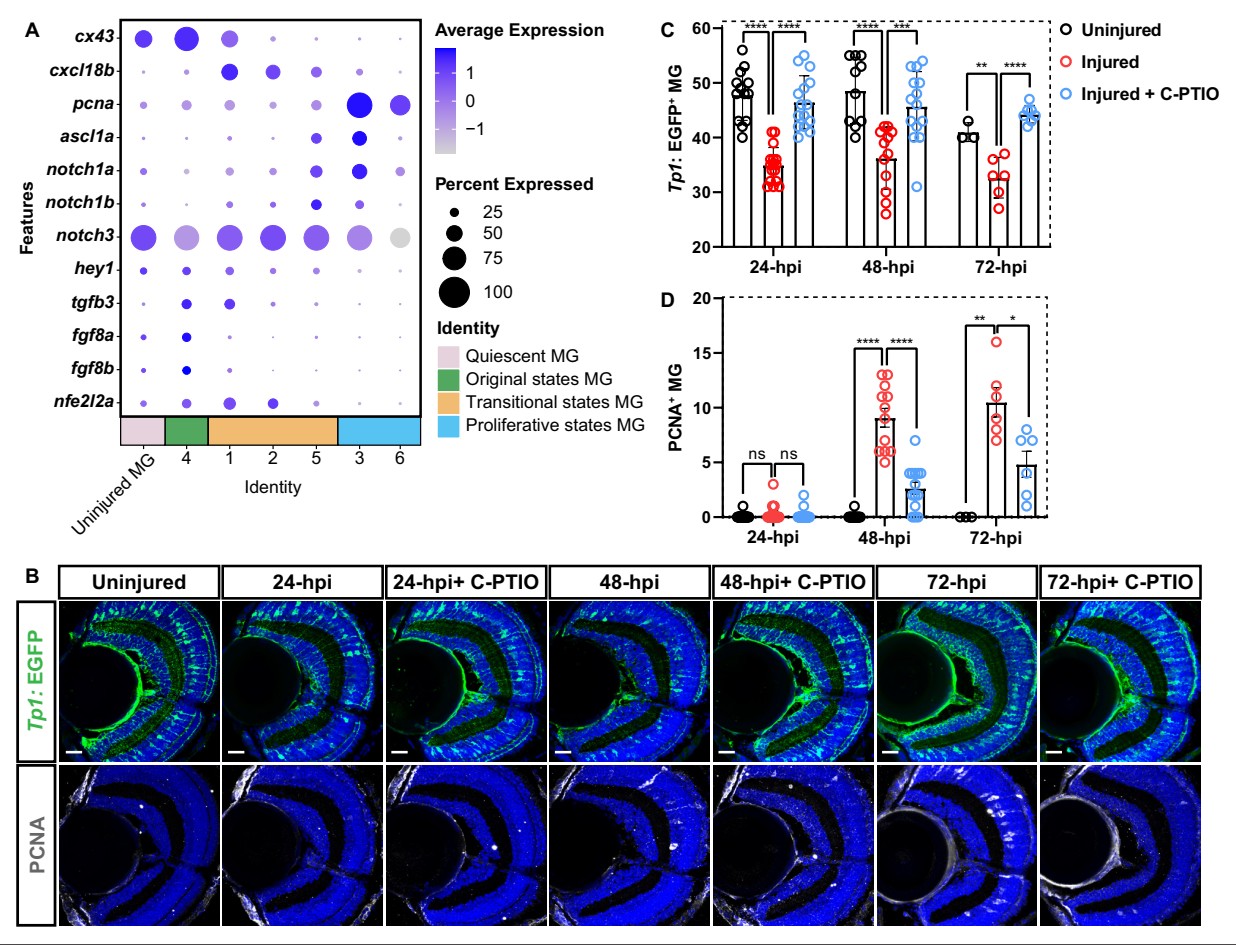

**Figure 7.** Nitric oxide (NO) regulates Müller glia (MG) proliferation by suppressing Notch signaling. (**A**) Dot plot showing the Notch signaling-related gene expression in different MG states. The average expression levels of these genes for all cells in each cluster are coded by the gray level. The percentage of cells expressing each gene within each cluster is coded by dot size. (**B**) Representative images showing the dynamic changes of Notch signaling activity (green, indicated by *Tg(Tp1: EGFP)*) and proliferative MG (white, PCNA⁺) following injury with nitric oxide (NO) blockade using C-PTIO. Scale bars: 20 μm. (**C–D**) Quantitative plots showing the number of Notch activation (*tp1*: GFP⁺ MG) in (**C**) and proliferative MG (PCNA⁺ MG) in (**D**) at different time points post-green/red (G/R) cone ablation. We collected uninjured retina (24 hr post-injury [hpi], n=4; 48 hpi, n=4; 72 hpi, n=3), injured retina (24 hpi, n=9; 48 hpi, n=4; 72 hpi, n=6), and retina treated with C-PTIO (24 hpi, n=6; 48 hpi, n=5; 72 hpi, n=7) (mean ± SEM; ****p<0.0001, ***p<0.001, **p<0.01, *p<0.05, ns, p>0.05; two-way ANOVA followed by Tukey's HSD test).

The online version of this article includes the following source data for figure 7:

**Source data 1.** Quantification of Notch-activated (*tp1*: GFP⁺) Müller glia (MG) and PCNA⁺ MG at different time points under uninjured, injured, and C-PTIO-treated conditions.

Notch signaling activation, which potentially drives MG to exit quiescence and enter proliferation. Together, our results highlight that NO signaling drove the injury-induced *cxcl18b*-defined transitional state MG to enter proliferation, which may be mediated by Notch pathway regulation.

## Discussion

Our study provides the single-cell transcriptomic landscape of MG state progression following the cone ablation at the larval stage. Combined with clonal analysis, we identified a previously unreported *cxcl18b*-defined transitional MG state as the essential routing for MG cell-cycle re-entry. It led to our further genetic analysis, revealing the cell type-specific and concentration-dependent regulatory mechanism of NO derived from *nos2b* underlying injury-induced MG proliferation. Furthermore, NO signaling accounted for a decreased Notch activity, which has been previously reported to be essential for MG proliferation after the injury. Finally, cell state-specific gene disruption revealed that

*cxcl18b*-defined MG transitional state-specific *nos2b* is required for injury-induced MG proliferation, which is a prime example of previously reported '5R' redox regulation (*Meng et al., 2021*). Thus, our study provides the novel precision redox mechanism underlying MG proliferation in response to cone ablation in the zebrafish retina, opening exciting possibilities for future research and potential therapeutic interventions.

## Identification of *cxcl18b*-defined MG transitional states following the cone ablation

Combined scRNA-seq analysis and clonal analysis, we identified a previously unreported transitional MG state, marked by the expression of *cxcl18b,* as the essential routing for MG to re-enter the cell cycle following the retinal injury (*Figure 1H*, *Figure 2F and G*). To our knowledge, it is the first transitional state verified by in vivo clonal analysis to show a faithful prediction of injury-induced MG proliferation. Notably, this *cxcl18b* induction in MG depends on microglial recruitment in response to cone ablation, suggesting this transitional state is an MG response to the signals derived from the inflammatory reaction (*Figure 2K*). The underlying mechanism of this crosstalk is crucial to be addressed. Interestingly, the *cxcl18b*-containing gene module was also expressed in the CMZ, a region crucial for adult retina neurogenesis for a lifetime (*Figure 3A–D*). However, its expression is mainly absent from the central regions of the developing retina (*Figure 3F*). It suggests that the *cxcl18b*-defined transitional state might represent a developmental state used by constitutive neuron generation programs and injury-induced neuron regeneration programs beyond embryonic development. Furthermore, the *cxcl18b*-defined transitional state exhibits robust redox-related characteristics, such as the expression of *sod1*, *sod2*, and *catalase* (*Figure 4—figure supplement 1A*). It led to the critical discovery of this study, which showed the essential role of *nos2b* in regulating injury-induced MG proliferation. Unfortunately, our preliminary effects on gene disruption using CRISPR/Cas9 in F0 founders did not observe a significant reduction in the number of proliferative MG after *cxcl18b* disruption (*Figure 2—figure supplement 1I*). However, a recent study reported the essential role of *cxcl18b* in heart regeneration using mutant fish, providing a mechanism of *cxcl18b* as innate immune signaling in injury-induced tissue regeneration (*Goumenaki et al., 2024*). Their results raised concern about the efficiency of *cxcl18b* disruption in our system. It is essential to use mutant fish to re-examine the role of *cxcl18b* in injury-induced MG proliferation in the future. Also, we cannot rule out the possibility that other co-factors are involved in the action of *cxcl18b* in MG regeneration, which is another critical issue that needs to be solved in the future.

## The possible mechanism of NO signaling underlying injury-induced MG proliferation

Our study, for the first time, demonstrated an essential role of NO signaling in regulating MG proliferation after the cone ablation. However, we still need to understand more about the underlying mechanism. There are two well-characterized molecular events responsible for MG proliferation following the retinal injury: a decreased Notch activity and an increased *ascl1α* expression (*Fausett et al., 2008*; *Jorstad et al., 2017*; *Ramachandran et al., 2010*). Previous studies have shown that Notch3 is responsible for this decreased Notch activity, leading to increased Ascl1a through the de-depression mechanism (*Campbell et al., 2021*; *Sahu et al., 2021*). Interestingly, unlike *notch3*, *notch1a* has been reported to stimulate MG proliferation (*Campbell et al., 2021*; *Campbell et al., 2022*). Our scRNA-seq analysis also showed that as MG progressed into the proliferative states after the cone ablation, *notch3* expression gradually declined (*Figure 7A*). In contrast, *ascl1α, notch1a, and notch1b* expressions were upregulated and peaked at the proliferative states (*Figure 7A*). Thus, both previous studies and our current analysis support the idea that the transcriptional regulation of Notch expression accounts for the decreased Notch activity after the injury. Intriguingly, NO has been reported to activate the Notch1 signaling cascade by promoting the release and accumulation of the Notch1 intracellular domain (NICD) through nitration reactions, subsequently enhancing tumorigenesis and stem-like features in various cellular systems (*Charles et al., 2010*; *López-Juárez et al., 2017*; *Villegas et al., 2018*). It raises the possibility that NO signaling regulates injury-induced MG proliferation through the posttranslational modification of Notch3. Previous studies have demonstrated two significant forms of NO-mediated post-modification: S-nitrosylation on cysteines and nitration on tyrosine (*Wu et al., 2014*). Our preliminary analysis showed that all 11 cysteine residues within the putative

γ-secretase-dependent cleavage sites are conserved between Notch1a and Notch3, while notable differences were observed at four tyrosine residues. It leads to an outstanding question of whether NO regulated injury-induced MG proliferation by decreasing Notch activity via tyrosine nitration of Notch3, which is worthwhile to elucidate.

### The production of NO signaling in MG following retina damage

Previous studies reported that *cxcl18b* is a reliable inflammatory marker (*Goumenaki et al., 2024*; *Torraca et al., 2017*), and different inflammation responses modulate MG proliferation in the damaged zebrafish retina (*Iribarne and Hyde, 2022*). The induction of *cxcl18b* may represent the inflammatory responses of MG after the cone ablation, pointing out the potential link between the inflammatory response and the emergence of NO signaling in MG. Previous studies have demonstrated that iNOS is induced in various tissues by proinflammatory cytokines (*Förstermann and Sessa, 2012*; *Pacher et al., 2007*). One of the approaches to test the role of inflammatory responses is to manipulate the levels of inflammatory responses in MG to see the NO production and MG proliferative behaviors. Also, previous studies appreciate the essential role of electrical activity in tissue regeneration (*Levin, 2009*; *Qin et al., 2023*); in particular, calcium signaling has been shown to regulate various molecular pathways for liver regeneration, including the hepatocyte growth factor-Met-tyrosine kinase (HGF-Met) transduction pathway (*Bedi et al., 2024*) and the epidermal growth factor receptor signaling (*Kimura et al., 2023*). It is interesting to speculate that the abnormal electrical activity of MG in the injured retina may result in an elevated level of intracellular calcium, which activates calmodulin and induces the conformation change of NOS to NO production (*Hanson et al., 2018*; *Jones et al., 2007*). A similar mechanism has been proposed in the long-term potentiation of excitatory post-synaptic structure (*Grover and Teyler, 1990*; *Kawamoto et al., 2012*; *Park, 2018*), as well as in the glutamate neurotoxicity model (*Ashpole et al., 2013*; *Ashpole et al., 2012*). Thus, the production of NO derived from *Nos* may be the product of the interplay between the inflammatory responses and the electrical activity in MG after the retina damage.

### Limitations of this study

A few limitations of this study should be acknowledged: (1) Our study focuses on larval zebrafish, in which the core features of MG and immune responses are conserved compared to the adult. However, we acknowledge that the adult retina—with its fully matured differentiated retina and immune response—provides irreplaceable biological insight. Nevertheless, larval models offer a powerful platform to uncover conserved regenerative mechanisms and serve as a valuable complement for identifying age-dependent differences in MG-mediated regeneration. (2) While our data suggest that injury-induced NO suppresses Notch signaling activation and promotes MG proliferation, the use of a general NO scavenger (C-PTIO) does not allow us to determine whether this regulation occurs in an autocrine or paracrine manner. The specific role of NO signaling within *cxcl18b*+ MG requires further validation using MG-specific NO depletion. (3) The current study's description of the landscape of MG transitional states is based on single-cell transcriptomic data obtained at a single time point (72 hpi) after the cone ablation, which may not provide a complete picture of the state transition of post-injury MG. Future scRNA-seq analysis of MG at multiple post-injury time points is necessary to clarify this issue. (4) While we demonstrated a critical role of NO in injury-induced MG proliferation, the potential contribution of microglia-derived NO was not directly examined. (5) We did not perform direct measurements of NO levels specifically within *cxcl18b*-defined MG cells, leaving open the question of localized NO signaling.

## Materials and methods

Detailed methods are provided in this version of this paper and include the following:

**Key resources table**

| Reagent type (species) or resource | Designation | Source or reference | Identifiers | Additional information |
|---|---|---|---|---|
| Strain, strain background (*Danio rerio*) | Wild type | Dr. William | ZIRC_ZL1 | AB |

*Continued on next page*

*Continued*

| Reagent type (species) or resource | Designation | Source or reference | Identifiers | Additional information |
|---|---|---|---|---|
| Strain, strain background (*Danio rerio*) | *lws2: nfsb-mCherry* | *Krylov et al., 2023* | ZDB-TGCONSTRCT-230530-2 | *Tg(opn1lws2: nfsb-mCherry)[uom3]* |
| Strain, strain background (*Danio rerio*) | *her4.1: dRFP* | *Yeo et al., 2007* | ZDB-TGCONSTRCT-070612-2 | *Tg(her4.1: dRFP)* |
| Strain, strain background (*Danio rerio*) | *gfap: EGFP* | *Bernardos and Raymond, 2006* | ZDB-FISH-150901-29,307 | *Tg(gfap: EGFP)* |
| Strain, strain background (*Danio rerio*) | *mpeg1: GFP* | *Ellett et al., 2011* | ZDB-TGCONSTRCT-170801–5 | *Tg(mpeg1: GFP)* |
| Strain, strain background (*Danio rerio*) | *Tp1bglob: EGFP* | *Yu and He, 2019* | ZDB-TGCONSTRCT-090625-1 | *Tg(Tp1bglob: EGFP)* |
| Strain, strain background (*Danio rerio*) | *pcna: GFP* | *Xu et al., 2020* | ZDB-LAB-070129-2 | *Tg(pcna: GFP)* |
| Strain, strain background (*Danio rerio*) | *ef1α: loxP-DsRed-loxP-EGFP* | This manuscript | | *Tg(ef1α: loxP-DsRed-loxP-EGFP)* |
| Strain, strain background (*Danio rerio*) | *cxcl18b: Cre-vmhc: ECFP* | This manuscript | | *Tg(cxcl18b: Cre-vmhc: ECFP; ef1α: loxP-DsRed-loxP-EGFP; lws2: nfsb-mCherry)* |
| Strain, strain background (*Danio rerio*) | *cxcl18b: GFP* | This manuscript | | *Tg(cxcl18b: GFP)* |
| Antibody | Mouse monoclonal anti-PCNA | Abcam | Cat#Ab29; RRID:AB_303394 | IF(1:500) |
| Antibody | Rabbit polyclonal anti-BLBP | Abcam | Cat#ab32423; RRID:AB_880078 | IF(1:1000) |
| Antibody | Mouse monoclonal anti-Glutamine Synthetase | BD Biosciences | Cat# 610518; RRID:AB_397880 | IF(1:1000) |
| Antibody | Chicken monoclonal anti-GFP | Abcam | Cat# ab13970; RRID:AB_300798 | IF(1:2000) |
| Antibody | Rabbit polyclonal anti-GFPtag | Rabbit polyclonal anti-GFPtag | Cat#50430–2-AP; RRID:AB_11042881 | IF(1:500) |
| Antibody | Rabbit polyclonal anti-DsRed2 | Takara Bio | Cat#632496; RRID:AB_10013483 | IF(1:1000) |
| Antibody | Alexa Fluor 488 AffiniPure Goat Anti-Mouse IgG (H+L) | Yeasen Biotech | Cat# 33206ES; RRID:AB_3662603 | IF(1:1000) |
| Antibody | Alexa Fluor 488 AffiniPure Donkey Anti-Rabbit IgG (H+L) | Yeasen Biotech | Cat# 34206ES60; RRID:AB_2909605 | IF(1:1000) |
| Antibody | Alexa Fluor 488 AffiniPure Donkey Anti-Chicken IgY (IgG) (H+L) | Jackson ImmunoResearch Labs | Cat# 703-545-155; RRID:AB_2340375 | IF(1:1000) |
| Antibody | Alexa Fluor 594 AffiniPure Donkey Anti-Mouse IgG (H+L) | Yeasen Biotech | Cat# 34112ES; RRID:AB_3661960 | IF(1:1000) |
| Antibody | Alexa Fluor 594 AffiniPure Goat Anti-Rabbit IgG (H+L) | Yeasen Biotech | Cat# 33112ES; RRID:AB_3661961 | IF(1:1000) |
| Antibody | Alexa Fluor 647-AffiniPure Goat Anti-Mouse IgG +IgM (H+L) | Jackson ImmunoResearch Labs | Cat# 115-605-044; RRID:AB_2338906 | IF(1:1000) |
| Recombinant DNA reagent | pTol2-cxcl18b: GFP | This manuscript | | We made this plasmid by ligating the cxcl18b promoter and GFP element |
| Recombinant DNA reagent | pTol2-cxcl18b: Cre-vmhc: mCherry | This manuscript | | We made this plasmid by ligating the cxcl18b promoter and Cre element |
| Recombinant DNA reagent | pTol2-cxcl18b: gal4FF | This manuscript | | We made this plasmid by ligating the cxcl18b promoter and gal4FF element |
| Recombinant DNA reagent | pUAS: Cas9T2ACre; U6: sgRNA1; U6: sgRNA2 | *Di Donato et al., 2016* | Addgene plasmid #74010; RRID:Addgene_74010 | |
| Recombinant DNA reagent | pTol2-UAS: Cas9-T2A-Cre-U6: nos2b sgRNA1; U6: nos2b sgRNA2 | This manuscript | | We made this plasmid by inserting two sgRNAs of *nos2b* in 10xUAS backbone |
| Commercial assay or kit | MEGAscriptTM T7 High Yield Transcription Kit | Invitrogen | Cat# AM1334 | |

*Continued on next page*

*Continued*

| Reagent type (species) or resource | Designation | Source or reference | Identifiers | Additional information |
|---|---|---|---|---|
| Commercial assay or kit | ClonExpress MultiS One Step Cloning Kit | Vazyme | Cat# C113-02 | |
| Commercial assay or kit | DIG RNA labeling kit | Roche | Cat# 11277073910 | |
| Chemical compound, drug | *N(ω)-nitro-L-arginine methyl ester* | Sigma-Aldrich | Cat# N5751-1G | 10 mM |
| Chemical compound, drug | *N(ω)-methyl-L-arginine acetate salt* | Sigma-Aldrich | Cat# M7033-5MG | 10 mM |
| Chemical compound, drug | 1400W dihydrochloride | MedChemExpress | Cat# HY-18730 | 200 nM |
| Chemical compound, drug | Phenyl-4,4,5,5-tetramethyl imidazoline-1-oxyl 3-oxide | Sigma-Aldrich | Cat# P5084-25MG | 10 mM |
| Chemical compound, drug | Dexamethasone | Sigma-Aldrich | Cat# D1756 | 10 mM |
| Chemical compound, drug | Metronidazole | Sigma-Aldrich | Cat# M3761-100G | 10 mM |
| Software, algorithm | FV10-ASW 4.0 Viewer | Olympus | https://olympus-global.com | Analysis image |
| Software, algorithm | GraphPad Prism V 9.0.0 | GraphPad Software | https://graphpad.com | Data analysis |
| Software, algorithm | Cell Ranger Single Cell Software Suite (v2.1.0) | 10x Genomics | https://support.10xgenomics.com | scRNA-seq data analysis |
| Software, algorithm | RStudio | RStudio IDE | https://posit.co/ | scRNA-seq data analysis |
| Software, algorithm | R 4.4.1 | R-project | https://www.r-project.org/ | scRNA-seq data analysis |
| Software, algorithm | Seurat | Satijalab | https://satijalab.org/seurat/ | scRNA-seq data analysis |

## Resource availability

### Lead contact
Further information and requests for resources and reagents should be directed to and will be fulfilled by the lead contact, Chang Chen (changchen@moon.ibp.ac.cn).

### Materials availability
All unique reagents generated in this study will be made available from the contact. We may require a completed MTA if there is potential for commercial application.

## Experimental model and subject details

### scRNA-seq data analysis
In this study, scRNA-seq raw data of MG enriched after G/R cone ablation at 72 hpi are from *Krylov et al., 2023*, embryonic RPCs at 24 hpf, and postembryonic RSCs at 14 dpf are from *Xu et al., 2020*. We re-processed the single-cell FASTQ sequencing reads (Novogene) and converted them to digital gene expression matrices using the Cell Ranger software (version 3.1.0) provided by 10x Genomics after mapping to the zebrafish GRCz11 (Ensembl release-96) genome assembly. An average of 47,545 mean reads per cell with 1343 median genes per cell in no ablation control, 63,177 mean reads per cell with 1556 median genes per cell in G/R cone ablation (lws2_72 hpi), 61,772 mean reads per cell with 2474 median genes per cell in the embryonic RPCs at 24 hpf, and 49,243 mean reads per cell with 1181 median genes per cell in the postembryonic RSCs at 14 dpf were obtained.

We then used the *cellranger aggr* (version 6.0, here) to aggregate the *cellranger* counts from the two datasets and normalize them to the same sequencing depth. The feature barcode matrix was recalculated and analyzed on the combined data. This resulted in a dataset of 19,706 aggregated cells, with 100% of the no ablation control reads retained and 71.4% of the *lws2_3 dpi* reads retained. The aggregated matrix for the no ablation control and lws2_72 hpi samples was loaded into the Seurat R package (version 4.3.0, https://satijalab.org/seurat/). Cells with gene expression of more than 200 and less than 4000, and with less than 5% mitochondrial content were filtered for further analysis. The filtered data were normalized, scaled, and clustered using principal component analysis with a significance threshold of $p<0.001$ (FindClusters, resolution = 0.5), and a UMAP was computed using scanpy.tl.umap, resulting in 20 distinct clusters. MG clusters were identified by high expression of marker genes, including *rlbp1a*, *fabp7a*, *slc1a2b*, *glula*, *glulb*, *gfap*, and *her4.1*. Proliferative MG

clusters were identified from proliferative cell clusters by high expression of proliferative cell markers, including *pcna*, *mki67*, *gfap*, *her4.1*, and low expression of the CMZ markers, including *fabp11a*, *col15α1b*, *fabp7b*, *rx2*. We further analyzed the MG and proliferative MG clusters to obtain 13 clusters, including no ablation control 5932 cells and G/R cone ablation 72 hpi 3999 cells in the UMAP plot (*Figure 1—figure supplement 1C*). We compared the proportion of cell numbers in each cluster before and after injury and chose clusters with a high proportion of cell numbers in G/R cone ablation 72 hpi (Clusters 2, 3, 5, 6, 9, 10, 11, 12, and 13) to further cluster into 10 clusters (*Figure 1—figure supplement 1D*).

scRNA-seq data of embryonic RPCs at 24 hpf and postembryonic RCSs at 14 dpf were generated into 10 and 11 clusters, respectively. Each cluster is identified by the cluster-specific marker gene and different development stage markers, including *fabp11a*, *col15α1b*, *cxcl18b*, *her4.2*, *npm1a*, *her9*, *fabp7a*, *dla*, *atoh7*, *vsx1*, *mafba*, *otx5*, and *rem1* (*Figure 3A and C* and *Figure 3—figure supplement 1A and B*).

## Pseudo-time trajectory analysis

After the UMAP cluster analysis of the single-cell data, trajectory analysis was performed to investigate the pseudo-time transcriptomic change of these 10 clusters using the 'monocle 2' R package, which revealed three distinct states (*Figure 1G*).

## Zebrafish husbandry and transgenic fish lines

All experimental zebrafish embryos, larvae, and adults were produced, grown, and maintained at 28°C according to standard protocols. Embryos and larvae were kept in embryo medium (E3; 5 mM NaCl, 0.17 mM KCl, 0.33 mM CaCl$_2$·2H$_2$O, 0.33 mM MgSO$_4$·7 H$_2$O, 1.3 × 10$^{-5}$% wt/vol methylene blue in RO water) at 28.5°C, under a 14:10 light:dark cycle. Animal procedures performed in this study were approved by the Animal Use Committee of the Institute of Neuroscience, Chinese Academy of Sciences (NA-069-2023).

Published fish lines used in this study include the following: AB (WT) and transgenic lines (Tg): *Tg(opn1lw2: NTR-mCherry)^{uom3}* (ZDB-ALT-201012-2) (*Wang et al., 2020*) in this study named *Tg(lws2: nfsb-mCherry)*, *Tg(gfap: EGFP)^{mi2001}* (ZDB-FISH-150901-29307) (*Bernardos and Raymond, 2006*), *Tg(her4.1: dRFP)* (ZDB-TGCONSTRCT-070612-2) (*Yeo et al., 2007*), *Tg(mpeg1: GFP)* (ZDB-TGCONSTRCT-170801–5) (*Ellett et al., 2011*), *Tg(pcna: GFP)* (ZDB-LAB-070129-2) (*Xu et al., 2020*), *Tg(ef1α: loxP-DsRed-loxP-EGFP)*(*Hans et al., 2009*), *Tg(Tp1bglob: EGFP)* (*Yu and He, 2019*).

Newly generated transgenic fish line contained: *Tg(cxcl18b: GFP)*, *Tg(cxcl18b: Cre-vmhc: mCherry:: ef1α: loxP-DsRed-loxP-EGFP:: lws2: nfsb-mCherry)*, and mutants *nos1^{+/-}*, *nos2a^{+/-}*, *nos2b^{+/-}*, *gsnor^{+/-}*, *nos1^{-/-}*, *nos2a^{-/-}*, *nos2b^{-/-}*, *gsnor^{-/-}*.

## Generation of *Tg(cxcl18b: GFP)*

The plasmid of *pTol2-cxcl18b: GFP* (10 ng/µl) and *tol2* mRNA (50 ng/µl) were co-injected into a WT embryo at the one-cell stage. Zebrafish embryos with green fluorescent were grown at the E3 medium according to standard protocol and selected as the founder (F0). The adult F0 fish crossed with the WT selected the fish with GFP$^+$ offspring as F1. In this stable transgenic line, *cxcl18b^+* cells expressed the fluorescent protein GFP, and the full name of this line is *Tg(cxcl18b: GFP)*.

## Generation of *Tg(cxcl18b: Cre-vmhc: mCherry:: ef1α: loxP-DsRed-loxP-EGFP:: lws2: nfsb-mCherry)*

The plasmid of *pTol2-cxcl18b: Cre-vmhc: mCherry* (10 ng/µl) and *tol2* mRNA (50 ng/µl) were co-injected into the embryo obtained from *Tg(lws2: nfsb-mCherry)* cross *Tg(ef1α: loxP-DsRed-loxP-EGFP)* at the one-cell stage. Zebrafish embryos were maintained at E3 medium according to standard protocol. Larvae with *mCherry* expression in the heart and eyes were selected and grown as the founder (F0) for further construction. The adult founder was crossed with WT (AB), considering the expression of *cxcl18b* during zebrafish development, larvae with mCherry and GFP double positive were selected and grown as the F1. In this zebrafish line, EGFP permanently marks all cells that have expressed *cxcl18b*, as well as their entire lineage of progeny. The full name of this line is *Tg(cxcl18b: Cre-vmhc: mCherry:: ef1α: loxP-DsRed-loxP-EGFP:: lws2: nfsb-mCherry)*.

## Generation of mutant fish lines

The CRISPR/Cas9 system was employed to efficiently and precisely generate mutant zebrafish lines (*Wu et al., 2018*). Two sgRNAs targeting gene coding sequences involved in NO metabolism signaling (NO synthase, *Nos*; S-nitrosoglutathione reductase, GSNOR) were designed based on the zebrafish GRCz11 genome assembly.

Primers were designed approximately 200 bp from the sgRNA target sites for genotyping (sgRNAs and genotyping primers are listed in *Supplementary file 1*). We mixed the two sgRNAs (100 ng/μl) for each target gene with Cas9 protein (400 ng/μl) and injected them into transgenic fish line *Tg(lws2: nfsb-mCherry)* embryos at the one-cell stage. Adult fish were crossed with WT, and offspring displaying *mCherry* expression in the eyes at 5 dpf were selected for further genotyping. Mutants were confirmed by sequencing, and the fish with open reading frame shift were collected as heterozygous fish for further studies.

## Plasmid construction

We cloned the *cxcl18b* regulatory element (3048 bp upstream of the start codon, including the *5'UTR*) as the *cxcl18b* promoter from the zebrafish genomic DNA (*Figure 2C*). The primers used for amplification were 5'-GCATTTGTCTCCTCATGCATTGACTAC-3' (forward primer) and 5'-TTGCTGCAAACT ATATGTAGGAAATGCTG-3' (reversed primer). For constructing the plasmids *pTol2-cxcl18b: GFP*, *pTol2-cxcl18b: Cre-vmhc: mCherry*, and *pTol2-cxcl18b: gal4FF*, each DNA elements cassette was inserted into the *pDestTol2pA2* vector (*Kwan et al., 2007*). The plasmid of *pUAS: Cas9T2AGFP; U6: sgRNA1; U6: sgRNA2* was kindly provided by Prof. Filippo Del Bene (*Di Donato et al., 2016*). To prepare for plasmid *pTol2-UAS: Cas9-T2A-Cre-U6-empty* and *pTol2-UAS: Cas9-T2A-Cre-U6: nos2b sgRNA1; U6: nos2b sgRNA2*, the same two *nos2b* sgRNAs used to construct *nos2b* mutant fish were inserted into *pUAS: Cas9T2AGFP; U6: sgRNA1; U6: sgRNA2* plasmid, with the *GFP* element replaced by *Cre* using ClonExpress MultiS One Step Cloning Kit (Vazyme, C113-02). For the plasmid construction of the MG-specific knockout system based on recombinant adenoviral vectors, regulatory elements were inserted into adenoviral vectors *pAdc68-S* according to the standard protocol outlined by *Jia et al., 2019*. The final plasmids generated included *pAdC68XY2-E1-CMV-GFP pAdC68XY2-E1-10xUAS: Cas9-2a-Cre-U6: sgRNA1(nos2b); U6: sgRNA2(nos2b), pAdC68XY2-E1-cxcl18b: gal4FF*, and *pAdC68XY2-E1-10xUAS: Cas9-2a-Cre-U6: empty* as the control. Plasmid construction primer sequences are listed in *Supplementary file 2*.

## MTZ treatment

Zebrafish larvae were exposed to a 10 mM MTZ solution (Sigma-Aldrich, M3761-100G) in standard fish water. For *Tg(lws2: nfsb-mCherry)* larvae processed for immunohistochemical analysis, MTZ treatment was initiated at 6 dpf to ablate green and red cones. Larvae were maintained in the MTZ solution at densities of fewer than 50 larvae per 50 ml Petri dish and kept at 28.5°C. The MTZ solution was refreshed every 24 hr to ensure continuous cone ablation until fixation in 4% paraformaldehyde (PFA), while control larvae were kept in standard fish water.

## Clone analysis of *cxcl18b* lineage-traced MG

Clone analysis of *cxcl18b*+ lineage-traced MG was restricted to cells located in the central and dorsal region of the zebrafish retina after G/R cone ablation in *Figures 2 and 6*, and their figure supplement. This spatial restriction strongly suggests that the proliferative MG originate from local mature MG, although we cannot completely rule out the possibility that CMZ-derived progenitors contribute to the generation of proliferative MG in the peripheral retina.

## Gene disruption via CRISPR/Cas9 system

We used the CRISPR/Cas9 system for efficient gene disruption. Two sgRNAs were designed to target the coding sequences of genes involved in the NO signaling pathway (including *nos1*, *nos2a*, *nos2b*, *gsnor*) and *cxcl18b*. The sgRNAs were in vitro transcribed and purified using the LiCl precipitation approach (MEGAscript T7 Transcription Kit, Invitrogen, AM1334). A mixture of the two sgRNAs (in total 200 ng/μl) with Cas9 protein (400 ng/μl, Novoprotein, E365-01A) was co-injected into the

embryo of *Tg(lws2: nfsb-mCherry)* at the one-cell stage. The efficiency of gene knockout for each sgRNA was validated, as presented in *Figure 4—figure supplement 1C*.

## Pharmacological treatment with NO scavenger and Dex or *Nos* inhibitors intraocular injection

Zebrafish larvae at 6 dpf were anesthetized in 0.04% MS222 (Sigma, A5040) for 30–45 s. Inhibitor solutions of 2 μl were prepared as follows: 10 mM $N(\omega)$-nitro-L-arginine methyl ester (L-NAME, Sigma-Aldrich, N5751-1G), a broad *Nos* inhibitor, diluted in sterile PBS from a 100 mM stock; 10 mM $N(\omega)$-methyl-L-arginine acetate salt (L-NMMA, Sigma-Aldrich, M7033-5MG), a specific inhibitor of *nNos*, diluted in sterile PBS from a 20 mM stock; and 200 nM 1400W dihydrochloride (SW1400, MedChemExpress, HY-18730), a specific inhibitor of *iNos*, diluted in sterile PBS from a 39.97 mM stock. PBS was used as the control. Each solution was loaded into glass capillaries prepared using a micropipette puller (Narishige, PC-10) and connected to a microinjector (Applied Scientific Instrumentation, MPPI-3). Intraocular injection of the inhibitor solutions was administered to the zebrafish eyes starting 2 days before G/R cone ablation and continued until 72 hpi (*Figure 4C*).

Additionally, zebrafish were pre-treated with 10 mM of the NO scavenger, phenyl-4,4,5,5-tetramethyl imidazoline-1-oxyl 3-oxide (C-PTIO, Sigma-Aldrich, P5084-25MG), diluted in standard water (from a 100 mM stock), or with Dex (Sigma-Aldrich, D1756), which was diluted in DMSO, starting 2 days before G/R cone ablation and continuing until 72 hpi (*Figures 2J and 4C*).

## The RT-qPCR after FACS-sorted MG

We crossed *Tg(lws2: nfsb-mCherry)* with *Tg(pcna: GFP)*, *Tg(her4.1: dRFP)*, *Tg(gfap: EGFP)*, and *Tg(cxcl18b: GFP)* to collect retina cell. The retinas were dissected and the cells dissociated for sorting and enrichment of signal-positive cells using fluorescence-activated cell sorting (FACS, Beckman, MoFlo XDP). After 72 hr post G/R cone ablation, GFP+/RFP+ cells were collected from the retinas of *Tg(lws2: nfsb-mCherry × pcna: GFP)* as the proliferative MG group. For the uninjured MG group, GFP+/RFP+ cells were collected from *Tg(lws2: nfsb-mCherry × gfap: EGFP × her4.1: dRFP)* uninjured retinas, while GFP+/RFP+ cells from injured retinas represented the 72 hpi MG group, and GFP-/RFP- cells were enriched as the other retinal cell types group in the 72 hpi retinas. Additionally, from *Tg(lws2: nfsb-mCherry × cxcl18b: GFP)* retinas, GFP+/RFP+ cells were collected as the *cxcl18b*+ MG group and single RFP+ cells as the G/R cone group at 72 hpi (*Figure 5A*).

Interesting cells were collected into a 1.5 ml tube containing lysis buffer (20 mg/ml PK in TE buffer) and total RNA was isolated using this lysis buffer. cDNA synthesis was conducted by adding RT mix containing: 200 U Superscript II reverse transcriptase (Invitrogen, 18064-014), 1 × First-strand buffer (Invitrogen, 18064-014), 5 mM DTT (Invitrogen, 18064-014), 20 U Recombinant RNase inhibitor (Clontech, 2313A), 6 mM $MgCl_2$ (Sigma, M8266), 1 μM TSO (*Picelli et al., 2013*); 8% PEG8000 (Sigma, P1458). PCR amplification was performed as previously described *Picelli et al., 2013*.

RT-qPCRs were carried out using TB Green Premix Ex Taq (Takara, RR420A) on a LightCycler 480 II real-time PCR detection system (Roche). The qPCR primers are listed in *Supplementary file 3*. The ΔΔCt method was used to determine the relative expression of mRNAs in different group retinae cells and normalized to *actin* mRNA levels. Each group comparison was performed using a two-way ANOVA followed by Tukey's HSD test. Error bars represented SEM. ****$p<0.0001$, ***$p<0.001$; **$p<0.01$; *$p<0.05$; ns, $p>0.05$.

## Construction and injection of adenoviral-based MG-specific knockout system in the zebrafish retina

The type of AdC68 used in our study is a replication-deficient chimpanzee adenovirus. The plasmids used for viral packaging were described above, and the amplification and purification of recombinant chimpanzee adenovirus followed the standard protocol (*Jia et al., 2019*; *Liu et al., 2021*). We generated four adenoviruses for this study, including *AdC68XY2-E1-CMV-GFP pAdC68XY2-E1-10xUAS: Cas9-2a-Cre-U6: sgRNA1(nos2b); U6: sgRNA2(nos2b)* (referred to as *UAS: Cas9-2a-Cre-sgnos2b*, infectious titer: $2.00 \times 10^{13}$ ifu/ml), *pAdC68XY2-E1-cxcl18b: gal4FF* (referred to as *cxcl18b: gal4*, infectious titer: $1.90 \times 10^{13}$ ifu/ml) and *pAdC68XY2-E1-10xUAS: Cas9-2a-Cre-U6: empty* (referred to as *UAS: Cas9-2a-Cre*, infectious titer: $1.75 \times 10^{13}$ ifu/ml) as a control.

All adenoviruses were diluted in sterile PBS to a final infectious titer: $1.00 \times 10^{13}$ ifu/ml. The *UAS: Cas9-2a-Cre-sgnos2b* and *cxcl18b: gal4* adenoviruses were mixed at a 1:1 ratio for the MG-specific knockout system, while the *UAS: Cas9-2a-Cre* with *cxcl18b: gal4* adenoviruses were similarly mixed as a control. To achieve the *cxcl18b*[+] MG cell-specific knockout *nos2b*, we performed the intraocular injection of the adenovirus mixtures into the eyes of *Tg(ef1α: loxP-DsRed-loxP-EGFP* x *lws2: nfsb-mCherry)* at 5 dpf. One day post-injection, the fish were treated with MTZ for 3 consecutive days to ablate the G/R cone (**Figure 6E**).

## Tissue preparation and immunostaining

Zebrafish were fixed in 4% PFA (Electron Microscopy Services, 157-8) overnight, then cryoprotected in 30% sucrose for 6 hr, flash-frozen, and cryosectioned at a thickness of 14 μm. Immunostaining was performed following the protocol described by *Tang et al., 2017*. The primary antibody, including mouse anti-PCNA (Abcam, ab29) at a 1:500 dilution, as well as mouse anti-GS (glutamine synthase, BD Transduction Laboratories, 610518), rabbit anti-BLBP (Abcam, ab32423), rabbit anti-DsRed (Clontech, 632496), rabbit anti-TaqGFP (Proteintech, 50430-2-AP), and chicken anti-GFP (Abcam, ab13970) each at a 1:1000 dilution. Secondary antibodies conjugated to Alexa Fluor 488, 594, or 647 (Jackson ImmunoResearch Laboratories Inc) were used at a 1:1000 dilution. Primary antibodies were incubated overnight at 4°C, while Alexa Fluor secondary antibodies were incubated at room temperature for 2 hr. DAPI staining was performed according to the standard protocol.

## In situ hybridization

In this study, three digoxigenin (DIG)-labeled RNA probes targeting endogenous *cxcl18b, fabp11a,* and *col15α1b* were synthesized using the MEGAscript T7 High Yield Transcription Kit (Invitrogen, AM1334) and the DIG RNA Labeling Kit (Roche, 11277073910), following the manufacturer's instructions. The cDNA of each gene was amplified by PCR using the following primers: *cxcl18b* -F: 5'-ATGGCATTCACACCCAAAGCG-3'; *cxcl18b* -R: 5'-TAATACGACTCACTATAGGGATTGGCCCTGCTGTTTTTGTG-3'; *fabp11a*-F: 5'-GTTGGAAACCGGACCAAACC-3'; *fabp11a*-R: 5'- TAATACGACTCACTATAGGGACGGCTCGTTGAGCTTGAAT-3'; *col15a1b* -F: 5'-CCTCAATGGAGGTCCTAAAGGT-3'; *col15α1b* -R: 5'-TAATACGACTCACTATAGGGACCAGCTTCTGAGACCAAGC-3'.

Following the in situ hybridization protocol described by *Tang et al., 2017*, fresh zebrafish retinal sections were incubated overnight in the HybEZ system (Advanced Cell Diagnostics, 310013) at 65°C with 200 ng of probe for each slide. The next day, slides were sequentially washed in 5× SSC buffer and incubated overnight at 4°C with an anti-DIG-POD antibody (Roche, 11093274910) diluted 1:500 in TNB buffer (TN buffer with 0.5% blocking reagent; Roche). On the third day, the signal was detected using the TSA Plus Cyanine 3 (PerkinElmer, NEL744001KT) or Cyanine 5 (PerkinElmer, NEL745001KT)/ Fluorescein System.

## Imaging

Images were taken using an inverted confocal microscope system (FV1200, Olympus) confocal microscope using 40× (silicon oil, 1.05 NA) or 60× (silicon oil, 1.3 NA) objectives.

## Quantifications and statistical analysis

All quantification and visualization were performed with FV31S-SW 2.3.1.163 Viewer (Olympus) and ImageJ. **Z intensity projection** was used to process the Z-stack images acquired from 14-μm-thick sections of zebrafish retina for statistical analysis. For cell counting, PCNA[+] MG and their lineage in this study were defined as one proliferative MG, *cxcl18b*[+] MG, and their lineage were counted as one *cxcl18b*[+] MG.

To perform the statistical analysis, p-values were calculated with GraphPad Prism 8 (or Microsoft Excel). The unpaired, non-parametric Wilcoxon test was applied for comparison of two groups. The one-way ANOVA, followed by Tukey's HSD test, was applied for comparison of different groups with one treatment. The two-way ANOVA, followed by Tukey's HSD test, was applied for comparison of four groups with two treatments. Error bars represent SEM. ****$p < 0.0001$, ***$p < 0.001$; **$p < 0.01$; *$p < 0.05$; ns, $p > 0.05$.

## Acknowledgements

This research was supported by grants from the National Key R&D Program of China (2022YFA1303000 to CC), the Chinese Academy of Sciences Strategic Priority Research Program B grants (XDB39000000 to CC), National Natural Science Foundation of China-Key program project (Grant No. 91849203 to CC), the Creative Research Groups of the National Natural Science Foundation of China (32321003 to JH), STI2030-Major Projects (2021ZD0204500 to JH), the National Key Research and Development Program of China (2020YFA0112700 to JH), National Natural Science Foundation of China (32471029 to JH).

## Additional information

### Funding

| Funder | Grant reference number | Author |
|---|---|---|
| National Key Research and Development Program of China | 2022YFA1303000 | Chang Chen |
| National Natural Science Foundation of China | 91849203 | Chang Chen |
| Chinese Academy of Sciences Strategic Priority Research Program B grants | XDB39000000 | Chang Chen |
| Creative Research Groups of National Natural Science Foundation of China | 32321003 | Jie He |
| STI2030-Major Projects | 2021ZD0204500 | Jie He |
| National Key Research and Development Program of China | 2020YFA0112700 | Jie He |
| National Natural Science Foundation of China | 32471029 | Jie He |

The funders had no role in study design, data collection and interpretation, or the decision to submit the work for publication.

### Author contributions

Aojun Ye, Conceptualization, Data curation, Formal analysis, Validation, Investigation, Visualization, Writing – original draft, Writing – review and editing; Shuguang Yu, Conceptualization, Data curation; Meng Du, Dongming Zhou, Resources, Methodology; Jie He, Conceptualization, Resources, Data curation, Supervision, Funding acquisition, Writing – original draft, Writing – review and editing; Chang Chen, Conceptualization, Funding acquisition, Validation, Visualization, Writing – original draft, Writing – review and editing

### Author ORCIDs

Aojun Ye ⓘ https://orcid.org/0009-0008-4702-5914
Shuguang Yu ⓘ https://orcid.org/0000-0001-6640-5420
Dongming Zhou ⓘ https://orcid.org/0000-0002-5989-7591
Jie He ⓘ https://orcid.org/0000-0002-2539-2616
Chang Chen ⓘ https://orcid.org/0000-0003-1008-1062

### Ethics

Animal Use Committee of the Institute of Neuroscience, Chinese Academy of Sciences (NA-069-2023).

Reviewer #1 (Public review): https://doi.org/10.7554/eLife.106274.3.sa1
Reviewer #2 (Public review): https://doi.org/10.7554/eLife.106274.3.sa2
Author response https://doi.org/10.7554/eLife.106274.3.sa3

## Additional files

### Supplementary files

Supplementary file 1. sgRNA sequences and genotyping primers, related to STAR methods.

Supplementary file 2. Plasmid construction primer sequences, related to STAR methods.

Supplementary file 3. qPCR primer sequences, related to STAR methods.

MDAR checklist

### Data availability

This paper does not report new raw scRNA-seq data and original code. Any additional information required to reanalyze the data reported in this paper is available from the lead contact upon request.

The following previously published datasets were used:

| Author(s) | Year | Dataset title | Dataset URL | Database and Identifier |
|---|---|---|---|---|
| Krylov A, Yu S, Newton A, He J, Jusuf PR | 2023 | Quiescent Müller glia heterogeneity influences regenerative response following photoreceptor ablation in the zebrafish retina | https://www.ncbi.nlm.nih.gov/geo/query/acc.cgi?acc=GSE218107 | NCBI Gene Expression Omnibus, GSE218107 |
| Xu B, Tang X, Zhang H, Du L, He J | 2020 | Unifying Developmental Programs for Embryonic and Post-Embryonic Neurogenesis in the Zebrafish Retina | https://www.ncbi.nlm.nih.gov/geo/query/acc.cgi?acc=GSE122680 | NCBI Gene Expression Omnibus, GSE122680 |

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
