## [Editor Report · eLife Assessment]

Following retinal injury, zebrafish Müller glia reenter the cell cycle and generate replacement cells; this potentially **valuable** study proposes that injury induces a *cxcl18b*+ transitional state in Müller cells, which then express nitric oxide, inhibiting Notch signaling and allowing Müller glial cells to reenter the cell cycle. However, the evidence supporting the claims is **incomplete**, and the authors have made interpretations and conclusions that are not supported by the data. Questions of the temporal expression and function of cxcl18b, as well as the source of potential inflammatory cues before cxcl18b expression, remain unanswered and technical limitations and data inconsistencies raise concerns. Using larval animals complicates the analysis since the retina is still forming, and distinguishing between injury-induced regeneration and ongoing development is complex. With more rigorous testing of the signaling pathways proposed and a clear demonstration of their interdependence, the link between nitric oxide signaling and Notch activity, particularly, would interest those investigating retinal regeneration.

---

## [Referee Report · Reviewer #1 (Public review)]

Summary:

This study presents a valuable contribution of NO signaling in zebrafish retinal regeneration in larval animals. The data on NO signaling are solid. There are multiple limitations to the study, but these are largely acknowledged by the authors in the revised text.

Strengths:

New data on NO signaling is valuable to the field but may be limited to larval "regeneration".

Weaknesses:

A weakness of the approach is testing cone ablation and regeneration in early larval animals. A near identical study was already done by Hoang et al 2020 in the adult zebrafish, a more relevant biological timepoint.

---

## [Referee Report · Reviewer #2 (Public review)]

Summary:

In this manuscript Ye at al. examine the sequence of events that occur in the damaged zebrafish Muller glia (MG) in states between quiescence and the onset of proliferation. Using an inducible metronidazole (MTZ) and nitroreductase system to ablate red/green cones in larval zebrafish, they identify a novel transitional MG state that is characterized by the expression of cxcl18b. Using trajectory analysis from single-cell RNA-seq datasets, they find that cxcl18b is expressed before MG expression PCNA and become proliferative. They find that cxcl18b expression peaks in MG at approximately 24 hours post injury (hpi) and rapidly declines as MG proliferate following injury. In a most interesting finding, the authors find a link between nos2b-dependent nitric oxide signaling and cxcl18b-mediated proliferation. Mutagenesis of nos2b decreases MG proliferation. The mechanism linking NO signaling to proliferation was suggested to function via notch signaling as pharmacological inhibition of nitric oxide signaling resulted in elevated Notch activity, thus preventing MG proliferation. The authors suggest a model whereby cxcl18b induces autocrine NO signaling in MG to reduce activity of Notch3, thereby promoting MG proliferation.

Strengths:

The authors utilize a number of sophisticated transgenic approaches and generate novel lines that will have value to the field. The identification of a novel cxcl18b transition state is exciting and the putative link between NO signaling and Notch activity would provide new insight into the drivers of Muller glia proliferation.

Weaknesses:

While the overall model is appealing and may serve as a foundation for future studies, some information gaps remain and certain conclusions rely on correlational data. The cellular expression of nos2b remains unclear as the single-cell RNA-seq data cannot provide expression data that matches RT-PCR results. The temporal sequence of events are based on transgene expression in the Tg(cxcl18b:GFP) lines, where persistence of the GFP fluorescence may not reflect endogenous cxcl18b. The identity of putative cxcl18b receptors on MG to support an autocrine signaling pathway remains unclear. Nevertheless, this is an interesting study that should open new avenues of exploration.

---

## [Author Response]

The following is the authors’ response to the original reviews.

**Reviewer #1:**
(1) The authors state that more is known about glial reactivation than cell-cycle re-entry. They are confusing many points here. More gene networks that require cell-cycle re-entry are known. Some of the genes listed for "reactivation" are, in fact, required for cell cycle re-entry/proliferation. And the authors confuse gliosis vs glial reactivation.

We thank the reviewer for this important and constructive comment. We fully agree that clearly distinguishing between the concepts of glial reactivation, glial proliferation, gliosis, and neurogenesis is essential to avoid conceptual confusion in our study.

Injury-induced retinal regeneration in zebrafish:

Glial reactivation refers to the initial response of quiescent Müller glia (MG) to injury, characterized by morphological changes and upregulation of reactive markers (e.g., gfap, ascl1a, lin28a) and activation of signaling pathways such as Notch, Jak/Stat, and Wnt (Lahne et al., 2020; Pollak et al., 2013; Sifuentes et al., 2016; Yao et al., 2016).

Glial proliferation refers to the clonal expansion of these MG-derived progenitor cells, which undergo rapid cell-cycle re-entry and amplify to generate sufficient progenitors for regeneration (Iribarne and Hyde, 2022; Lee et al., 2024; Wan and Goldman, 2016)

Gliosis vs neurogenesis represents a divergent fate decision following proliferation. In zebrafish, MG-derived progenitor cells differentiate into retinal neurons that can replace those damaged or lost due to retinal injury. In contrast, mammalian MG tend to undergo an initial gliotic surge and rapidly revert to a quiescent state, exhibiting gliosis and glial scarring (Thomas et al., 2016; Yin et al., 2024). Thus, we totally agreed that gliosis cannot be confused with glial reactivation because glial reactivation is the very first step of glial injury responses, whereas gliogensis is the very last glial response to the injury.

We agree with the reviewer that many genes typically described as “reactivation markers” (e.g., ascl1a, lin28a, sox2, mycb, mych) are also essential regulators of cell-cycle re-entry (Gorsuch et al., 2017; Hamon et al., 2019; Lee et al., 2024; Lourenço et al., 2021; Pollak et al., 2013; Thomas et al., 2016). Because the glial reactivation is a leading event for glial proliferation, the regulators of glial reactivation are expected to be responsible for glial proliferation as well.

In our study, we focused on the states preceding glial proliferation to understand the mechanism underlying injury-induced glial cell-cycle re-entry. We defined these transitional states and the subsequent proliferative MG states based on single-cell RNA-seq trajectory analysis. (revised lines: 41-58)

(2) A major weakness of the approach is testing cone ablation and regeneration in early larval animals. For example, cones are ablated starting the day that they are born. MG that are responding are also very young, less than 48 hrs old. It is also unclear whether the immune response of microglia is a mature response. All of these assays would be of higher significance if they were performed in the context of a mature, fully differentiated, adult retina. All analysis in the paper is negatively affected by this biological variable.

We thank the reviewer for raising this important point regarding the developmental stage of the retina in our model system. We have carefully considered this concern and now provide additional clarification and justification, as follows:

(1) The glial responses in larval and adult retina:

Previous studies have demonstrated that injury-induced glial responses are largely conserved in larval and adult zebrafish retina, including reactive gliosis marked by gfap upregulation and proliferation(Meyers et al., 2012; Sarich et al., 2025). In our study, G/R cones were ablated beginning at 5 dpf using metronidazole (MTZ), and we observed robust induction of PCNA⁺ MG in the inner nuclear layer, consistent with injury-induced proliferation (Figure 1E). These findings align with previous studies showing that key features of MG regenerative responses are conserved across larval and adult stages.

(2) The microglial responses in larval and adult retina:

Retinal microglia functionally mature at 5 dpf in the zebrafish retina (Mazzolini et al., 2020; Svahn et al., 2013), and prior studies have demonstrated that microglia in larval and adult zebrafish exhibit similar responses to injury, including migration, morphological activation, and phagocytosis(Nagashima and Hitchcock, 2021; White et al., 2017). In our experiments using Tg(mpeg1: GFP) larvae, we observed clear microglial recruitment to the outer nuclear layer (ONL) following cone ablation (Figure 1E and Figure 1-figure supplement 1A), supporting the functional competence of larval microglia in injury-induced immune responses

(3) The contribution using larval animals to study the regeneration program:

We agree that regeneration studies in the adult retina can provide important biological insights, particularly in a fully differentiated tissue environment. Accordingly, we have acknowledged this limitation in our revised manuscript “limitations of this study” section (revised lines 534-540: “1. Our study focuses on larval zebrafish, in which the core features of MG and immune responses are conserved compared to the adult. However, we acknowledge that the adult retina—with its fully matured differentiated retina and immune response—provides irreplaceable biological insight. Nevertheless, larval models offer a powerful platform to uncover conserved regenerative mechanisms and serve as a valuable complement for identifying age-dependent differences in MG-mediated regeneration.”) and have stated our intention to extend future analyses to adult zebrafish, especially to explore age-dependent differences in redox signaling and MG proliferation. At the same time, we believe that the larval model offers unique advantages for uncovering fundamental, conserved mechanisms of regeneration and enables characterization of age-dependent regulatory differences. Thus, our study in larval animals serves as a complementary and informative platform for understanding both the conserved and developmental stage-specific features of injury-induced regeneration.

(4) Related to the above point, the clonal analysis of cxcl18b+ MG is complicated by the fact that new MG are still being born in the CMZ (as are new cones for that matter).

We thank the reviewer for raising this important point regarding potential contributions from CMZ-derived progenitors to the lineage-traced cxcl18b⁺ MG clones. To address this concern, we have implemented evidence to rule out a CMZ origin for the clones analyzed:

Spatial restriction of clones: All clones included in our analysis were located exclusively within the central and dorsal retina, as shown in Figure 2H. From the spatial distribution of reactive MG populations across the retina, we observed a patterned organization in which the vast majority of proliferating MG arose from local mature MG–derived progenitors, rather than from peripheral CMZ-derived progenitors. However, we acknowledge that we cannot entirely exclude the possibility that CMZ-derived progenitors contribute to injury-induced MG proliferation, particularly in the peripheral retina.

We have clarified this point in the revised Methods section (revised lines 756–762: “Clone analysis of cxcl18b^+^ lineage-traced MG was restricted to cells located in the central and dorsal region of the zebrafish retina after G/R cone ablation in Figure 2, Figure 6, and their figure supplement. This spatial restriction strongly suggests that the proliferative MG originate from local mature MG, although we cannot completely rule out the possibility that CMZ-derived progenitors contribute to the generation of proliferative MG in the peripheral retina.”) and updated the corresponding figure legends.

(4) A near identical study was already done by Hoang et al., 2020, in adult zebrafish, a more relevant biological timepoint. Did the authors check this published RNA-seq database for their gene(s) of interest?

We thank the reviewer for pointing out the relevance of the study by Hoang et al., 2020, which characterized the transcriptional dynamics of MG reactivation in the adult zebrafish retina. We agree that comparisons with their single-cell RNA-seq dataset are important to confirm the conservation of our findings in larval vs adult zebrafish.

To this end, we examined the adult zebrafish MG dataset reported by Hoang et al., and confirmed that cxcl18b is also present and enriched in their analysis, particularly in activated MG populations under various injury paradigms:

(1) cxcl18b is listed as a differentially expressed gene (DEG) in Supplementary Table ST2, enriched in GFP⁺ MG following injury. It is also significantly upregulated in both NMDA-induced and light damage conditions, as shown in Supplementary Table ST3.

(2) In Supplementary Table ST5, cxcl18b is identified as a classifier of activated MG, with classification power scores of 0.552 (NMDA), 0.632 (light damage), and 0.574 (TNFα + γ-secretase inhibitor treatment), indicating its consistent expression across multiple injury models.

(3). In their pseudotime analysis (Figure 4C and Supplementary Table ST8), cxcl18b is specifically expressed in Module 5, which is expressed earlier along the trajectory than ascl1a. This temporal pattern of cxcl18b preceding ascl1a expression is consistent with our trajectory analysis in larval MG (Figure 1H), further supporting its role as an early marker of the transitional state before proliferation.

These findings underscore the robustness and biological relevance of cxcl18b as a conserved marker of injury-responsive MG in both larval and adult zebrafish. Our data expand upon the prior work by specifically characterizing a cxcl18b-defined transitional MG state preceding cell-cycle re-entry, thereby offering additional insights into the temporal staging of MG activation during regeneration.

(5) KD of cxcl18b did not affect MG proliferation or any other defined outcome. And yet the authors continually state such phrases as "microglia-mediated inflammation is critical for activating the cxcl18b-defined transitional states that drive MG proliferation." This is false. Cxcl18b does not drive MG proliferation at all.

We thank the reviewer for raising this concern. We agree with the reviewer and have revised this statement as "These findings suggest that microglia-mediated inflammation may contribute to the activation of cxcl18b-defined transitional states that precede MG proliferation, although a causal relationship remains to be established." (revised lines 251-253).

(6) A technical concern is that intravitreal injections are not routinely performed in larval fish.

We appreciate the reviewer’s technical concern regarding the use of intravitreal injections in larval zebrafish. In our study, we performed intraocular injection according to previously established methods (Alvarez et al., 2009; Giannaccini et al., 2018; Rosa et al., 2023). This approach involves carefully delivering a small volume of viral suspension into the intraocular space by a glass micropipette. To address this concern, we will revise the Materials and Methods section to clearly describe the injection procedure and will cite the relevant references accordingly.

**Reviewer #2:**
(1) The authors note a peak of PCNA+ Muller glia at 72 hours post injury. This is somewhat surprising as the MG would be expected to generate progenitor cells that would continue proliferating and stain with PCNA. Indeed, only a handful of PCNA+ cells are seen in the INL/ONL layer in Figure 1E2 with few clusters of progenitors present. It would be helpful to stain with a Muller glia marker to confirm these PCNA+ cells are Muller glia. It's also curious that almost all the PCNA+ cells are in the dorsal retina, even though G/R cone loss extends across both dorsal and ventral retina. Is this typical for cone ablation models in larval zebrafish?

We thank the reviewer for their insightful comment regarding the spatial distribution and identity of PCNA⁺ cells following injury.

In our study, we observed that the injury-induced proliferating cells (PCNA⁺) were predominantly located in the central and dorsal regions of the retina at 72 hours post-injury (hpi) (Figure 1E). To verify the identity of these proliferating cells, we performed additional immunostaining using BLBP, and confirmed that the majority of PCNA⁺ cells also express BLBP (Figure 1–figure supplement 1B in our revised Data), these results supporting their MG origin.

The regional bias of MG proliferation towards the central and dorsal retina is consistent with previous findings. Notably, (Krylov et al., 2023) demonstrated that MG exhibit region-specific heterogeneity in their regenerative responses to photoreceptor ablation. Their study identified proliferative MG subpopulations predominantly in the central (fgf24-expressing) and dorsal (efnb2a-expressing) domains, whereas ventral MG showed limited proliferative capacity (Krylov et al., 2023). These observations provide a plausible explanation for the spatially restricted PCNA⁺ MG population observed in our model following cone ablation.

(2) In Line 148: What is meant by "most original MG states" in this context? Original meaning novel? Or original meaning the earliest state MG adopted following injury? The language here is confusing.

We thank the reviewer for pointing out the ambiguous phrasing in our original manuscript. The term “most original MG states” was imprecise and misleading, as it could be interpreted as referring to the quiescent state of MG. In our context, we intended to describe the earliest transitional states in MG respond to injury, as they begin to exit quiescence and enter reactive characteristics. These early transitional MG populations co-express quiescent markers such as cx43 and early reactive markers gfap, as shown in Figure 1H.

To avoid confusion and improve conceptual clarity, we have revised the manuscript by replacing “most original MG states” with “early transitional MG state” (revised line 154) and have added a clearer explanation in the corresponding Results section to define this population more accurately.

(3) Perhaps provide a better image in Figure 2A of the cxcl18b at 48 hpi and 72 hpi. The current images appear virtually identical, with very little cxcl18b expression observed, especially compared to the 24 hpi. This is in contrast to the Tg(cxcl18b:GFP) transgenic line shown in Figure 2D, which indicates either much higher expression in proliferating cells at 48 hpi or the stability of GFP protein. Can the authors provide guidance on the accurate temporal expression of cxcl18b? Does expression peak rapidly at 24 hpi and then rapidly decline or is there persistence of expression to 48-72 hpi?

We appreciate the reviewer’s careful observation regarding the apparent similarity of cxcl18b expression at 48 hpi and 72 hpi in the in situ hybridization (ISH) images (Figure 2A), and the differences compared to the Tg(cxcl18b: GFP) reporter line shown in Figure 2D.

(1) The similarity of ISH images at the 48 hpi and 72 hpi (Figure 2A):

The cxcl18b mRNA signal peaked at 24 hpi, suggesting a rapid transcriptional response after retina injury. By 48 hpi, cxcl18b expression had already declined substantially, and by 72 hpi, the signal was further reduced to near-background levels. This temporal expression pattern explains why the ISH images at 48 hpi and 72 hpi appear nearly identical and much weaker compared to 24 hpi.

(2) The discrepancy between ISH and GFP reporter signal (Figure 2D):

The Tg(cxcl18b: GFP) reporter line shows persistent GFP expression beyond the transcriptional window of cxcl18b mRNA. This may be due to the prolonged stay of GFP protein, which remains detectable even after the endogenous transcription of cxcl18b has diminished. This explanation is also noted in the manuscript (revised lines 198–200). As a result, GFP⁺ MG cells are still visible at 48–72 hpi, and some of them co-label with PCNA.

These findings are consistent with our Pseudotime analysis based on scRNA-seq data (Figure 1H), which shows that cxcl18b expression precedes the induction of proliferative markers such as pcna and ascl1a.

(4) Line 198: The establishment of the Tg(cxcl18b:Cre-vhmc:mcherry::ef1a:loxP-dsRed-loxP-EGFP::lws2:nfsb-mCherry) is considerable but the nomenclature doesn't properly fit. Is the mCherry fused with Cre and driven by the cxcl18b promoter? What is the vhmc component? Finally, while this may provide the ability to clonally track cxcl18b-expressing MG, it does not address the prior question of what is the actual temporal expression of cxcl18b? If anything, this only addresses whether proliferating MG expressed cxcl18b at some point in their history, but does not indicate that cxcl18b expression co-exists in proliferating cells. The most convincing evidence is in Supplemental Figure 2B.

The "vmhc" component refers to the ventricular myosin heavy chain promoter, commonly used to label atrial cardiomyocytes (Jin et al., 2009). We cloned the vmhc upstream region containing its promoter and fusing with mCherry for selection during transgenic fish line construction.

Clone analysis using the Tg(cxcl18b: Cre-vmhc: mCherry::ef1a: loxP-DsRed-loxP-EGFP::lws2: nfsb-mCherry) further indicates that cxcl18b-defined the transitional state is the essential routing for MG proliferation. We have clarified in the revised text that this lineage tracing indicates a “history of injury-induced cxcl18b expression” rather than its ongoing expression during proliferation (revised line 205).

(5) Line 203: The data shown in Figure 2F do not indicate that these MG are cxcl18b+. Rather, the data are consistent with the interpretation that these MG expressed Cre at some prior stage and now express GFP from the ef1a promoter rather than DsRed. Whether these MG continue to express cxcl18b at the time these fish were collected is not addressed by these data. It is not accurate to conclude that these cells are cxcl18b+.

We thank the reviewer for pointing out this important issue. We agreed that the GFP^+^ MG shown in Figure 2F represents cells that have previously expressed cxcl18b and thus belong to the cxcl18b-expressing cell lineage, but this does not indicate that they continue to express cxcl18b at the time of sample collection. Performing clonal analysis using the Cre-loxp system, the GFP signal reflects historical cxcl18b promoter activity rather than ongoing transcription. We have revised the relevant sentence in our manuscript to clarify this point and now refer to these GFP^+^ cells as "cxcl18b lineage-traced MG" rather than "cxcl18b^+^ MG" to avoid any misinterpretation (revised line 207).

(6) Line 213: The statement that proliferative MG mostly originated from cxcl18b+ MG transitional states is a conclusion that does appear fully supported by the data. Whether those MG continue to express cxcl18b remains unanswered by the data in Figure 2 and would likely be inconsistent with the single-cell data in Figure 1.

We thank the reviewer for this valuable comment. We agree that the original statement on Line 213 regarding the lineage relationship between cxcl18b⁺ transitional MG and proliferative MG required clarification.

(1) The cxcl18b expression dynamics:

Our single-cell RNA-seq and ISH analyses consistently show that cxcl18b expression peaks as early as 24 hpi and declines rapidly, with significantly reduced expression by 48 and 72 hpi. These findings suggest that cxcl18b marks an early transitional MG state, rather than being maintained in proliferative MG. Indeed, in our scRNA-seq pseudotime trajectory analysis (Figure 1H), cxcl18b expression is highest in early transitional MG clusters (Clusters 1) and downregulated as cells progress toward proliferative states (Clusters 3/6), supporting a model in which cxcl18b is downregulated before cell-cycle re-entry.

(2) Prolonged stability of GFP protein:

The GFP signal observed in Tg(cxcl18b: GFP) retinas at 72 hpi may be because of the prolonged stability of GFP protein, rather than sustained cxcl18b transcription. The actual expression dynamics of cxcl18b are more directly reflected by our in situ hybridization and single-cell RNA-seq data, both showing a rapid decline after its early peak at 24 hpi. This explanation is also noted in the manuscript (revised lines 196–197).

(7) Line 246: The use of Dexamethasone to block inflammation is a widely used approach. However, dexamethasone is a broad-spectrum anti-inflammatory molecule that works through glucocorticoid signaling that may involve more than microglia. The observation that microglia recruitment and cxcl18a expression are both reduced is correlative but does not prove causation. Thus, the data are not sufficient to conclude that microglia-mediated inflammation is critical for activating cxcl18b expression. Indeed, data in Figure 1 indicate that cxcl18b expression occurs prior to microglia migration to the ONL.

We thank the reviewer for this thoughtful and important comment. We fully acknowledge that dexamethasone is a broad-spectrum anti-inflammatory agent that acts via glucocorticoid receptor signaling and may influence multiple immune and non-immune pathways beyond microglia.

In our study, dexamethasone treatment led to a reduction in both microglial recruitment and the number of cxcl18b^+^ MG at 72 hpi, suggesting a potential association between inflammation and cxcl18b activation. However, we agree that this observation remains correlative and is not sufficient to establish a direct link between microglia activity and cxcl18b induction. Our time-course analysis indicates that cxcl18b expression peaks at 24 hpi, preceding robust microglial accumulation in the ONL, further highlighting the need to clarify the temporal dynamics and cellular sources of inflammatory cues.

To address this question more conclusively, selective ablation of microglia during cone injury would be necessary. However, implementing such an approach would require a complex intersection of three transgenic lines—Tg(mpeg1: nfsB-mCherry) for microglia ablation, Tg(lws2: nfsB-mCherry) for cone ablation, and Tg(cxcl18b: GFP) for reporting—posing substantial genetic and experimental challenges.

We have revised the Results section accordingly to state: “These findings suggest that microglia-mediated inflammation may contribute to the activation of cxcl18b-defined transitional states that precede MG proliferation, although a causal relationship remains to be established.” (revised lines 251–253). We also added a new paragraph in the “Result: Clonal analysis reveals injury-induced MG proliferation via cxcl18b-defined transitional states associated with inflammation” as “While dexamethasone suppressed both microglial recruitment and cxcl18b^+^ MG generation, its broad anti-inflammatory action precludes definitive conclusions about microglial causality. Dissecting this relationship would require concurrent ablation of microglia and cone photoreceptors using a triple-transgenic strategy, which is beyond the scope of the current study. Targeted approaches will be necessary to resolve the specific role of microglia in initiating cxcl18b expression.” (revised lines 251–258) to explicitly acknowledge this limitation and the need for future studies using microglia-specific ablation models to resolve the mechanism.

(8) Could the authors clarify the basis of investigating NO signaling, given the relative expression of the genes by either cxcl18b+ MG or uninjured MG? Based on the expression illustrated in Supplemental Figure 4A, there is almost no expression of nos1 or nos2b in any MG. The authors are encouraged to revisit the earlier single-cell data sets to identify those cells that express components of NO signaling to determine the source(s) of NO that could be impacting the Muller glia.

We thank the reviewer for raising these important points.

Nitric oxide (NO) signaling has been implicated in the regeneration of multiple zebrafish tissues, including the heart (Rochon et al., 2020; Yu et al., 2024), spinal cord (Bradley et al., 2010), and fin (Matrone et al., 2021). Based on these findings, we hypothesized that NO signaling might also contribute to retinal regeneration.

As described in the manuscript, we compiled a redox-related gene list and systematically screened their roles in injury-induced MG proliferation using CRISPR-Cas9-mediated gene disruption. Among the candidates, disruption of nos genes significantly reduced the number of PCNA^+^ MG cells following G/R cone ablation (Figure 4), prompting us to further investigate the role of NO signaling.

(9) Line 319-320: this sentence appears to be missing text as "while no influenced across the nos mutants and gsnor mutants" does not make sense.

We appreciate the reviewer’s observation and agree that the original sentence was unclear. We have revised the sentence in the manuscript as follows:

“In contrast, no significant change in MG proliferation was observed in nos1, nos2a, or gsnor mutants compared to wild type (Figures 4F–4I)” (revised lines 326-328).

(10) Line 326-328: The text should be rewritten as the current meaning would suggest there was no significant loss of photoreceptors in the nos2b mutants. That is incorrect. Rather, there was no significant difference between WT and the nos2b mutants in the number of photoreceptors lost at 72 hpi following MTZ treatment. Both groups lost photoreceptors, but the number lost in nos2b hets and homozygotes was the same as WT.

We agree with the suggestion and have revised our manuscript. We have revised the sentence in the manuscript as follows:

“We observed no significant difference in the loss of cone photoreceptor at 72 hpi between nos2b mutants and WT, indicating that the reduced MG proliferation observed in nos2b mutants is independent of the injury (WT: 45 ± 8 remaining cones, n = 24; nos2b⁺/⁻: 49 ± 12, n = 20; nos2b⁻/⁻: 46 ± 9, n = 20; mean ± SEM) (Figure 4K).” (revised lines 331-335).

(11) There is concern over the inconsistencies with some of the data. In Figure 4, Supplement 1A, the single-cell data found virtually no expression of nos2b in either uninjured MG or cxcl18b+ MG. In contrast, the authors find nos2b expression by RT-PCR in the cxcl18b:GFP+ MG. The in situ expression of nos2b in Figure 5 - Supplement 1 is not persuasive. The red puncta are seen in a single cxcl18b:GFP+ cell but also in the plexiform layer and is other non cxcl18b:GFP+ cells.

We appreciate the concern regarding the apparent inconsistencies in nos2b expression across different datasets. We provide the following explanations:

(1) Low expression of nos2b in scRNA-seq data:

We propose a potential explanation: Nitric oxide (NO) signaling is known to exert its biological functions in a dose-dependent manner and is tightly regulated post-transcriptionally, especially in inducible nitric oxide synthase (iNOS) (Bogdan, 2001; Nathan and Xie, 1994; Thomas et al., 2008). Thus, even modest changes in nos2b expression may exert meaningful biological effects without producing strong transcriptional signals detectable by scRNA-seq, which could fall below the detection threshold of scRNA-seq methods. Supporting this idea, our functional assay (Figure 4J) reveals a clear concentration-dependent effect of NO on MG proliferation, consistent with the biological relevance of Nos2b activity despite its low transcript abundance.

(2) Regarding the in situ hybridization data:

We used both commercially available in situ hybridization probes from (HCR) and RNAscope (data not shown) to detect nos2b transcripts. While the nos2b signal was observed in other retinal cell types, including cells in the plexiform layer, our primary study was focused on examining its expression within the cxcl18b^+^ MG lineage.

(3) Regarding RT-PCR detection of nos2b in cxcl18b: GFP^+^ MG:

To enhance detection sensitivity, we enriched cxcl18b: GFP^+^ MG by FACS at 72 hpi and performed cDNA amplification before RT-PCR. This approach allowed the detection of low-abundance transcripts such as nos2b. It is also important to note that RT-PCR reflects fold changes in expression compared to MG to other retina cell type. The subtle but biologically upregulated of nos2b expression may not be readily captured by in situ hybridization or scRNA-seq.

(12) Line 356 - there is a disagreement over the interpretation of the current data. The statement that nos2b was specifically expressed in cxcl18b+ transitional MG states is not entirely accurate. This conclusion is based on expression of GFP from a cxcl18b promoter, which may reflect persistence of the GFP protein and not evidence of cxcl18b expression. Even assuming that the nos2b in situ hybridization and RT-PCR data are correct, the data would indicate that nos2b is expressed in proliferating MG that are derived from the cxcl18b+ transitional states. The single-cell trajectory analysis in Figure 2 indicates that cxcl18b is not co-expressed with PCNA. Furthermore, the single-cell data in Figure 4, Supplement 1, indicates no expression of nos2b in cxcl18b+ MG. The authors need to reconcile these seemingly contradictory pieces of data.

We thank the reviewer for this thoughtful and important comment. We agree that clarification is needed to accurately interpret the relationship between cxcl18b, nos2b, and MG proliferation, particularly considering the different temporal and technical contexts of our datasets.

(1) Lineage labeling and interpretation of GFP expression:

We acknowledge that in the Tg(cxcl18b: Cre-vhmc: mcherry::ef1a: loxP-dsRed-loxP-EGFP::lws2: nfsb-mCherry) line, GFP expression reflects historical activity of the cxcl18b promoter, rather than ongoing transcription. This GFP signal, due to its prolonged stay, may persist beyond the time window of endogenous cxcl18b expression. Accordingly, we have revised the manuscript to replace “cxcl18b⁺ MG” with “cxcl18b⁺ lineage-traced MG” throughout the relevant sections to prevent potential misinterpretation.

(2) Functional experiments support a lineage relationship between cxcl18b⁺ states and nos2b activity:

To further investigate the regulatory relationship between cxcl18b and nos2b, we conducted NO scavenging experiments using C-PTIO in the Tg(cxcl18b: GFP) background. We observed that the generation of cxcl18b: GFP⁺ MG after injury was not affected by NO depletion, indicating that cxcl18b activation precedes NO signaling (data not shown). However, PCNA⁺ MG was significantly reduced under the same treatment, suggesting that NO signaling is not required for cxcl18b⁺ transitional state formation, but is necessary for proliferation. Together with our MG-specific nos2b knockout data, these results support a model in which nos2b-derived NO acts downstream of the cxcl18b⁺ transitional state to promote MG cell-cycle re-entry.

(3) The scRNA-seq data with nos2b expression:

We agree with the reviewer that our scRNA-seq dataset shows minimal overlap between cxcl18b and pcna expression, which is consistent with our interpretation that cxcl18b expression marks a transitional phase before cell-cycle entry. Furthermore, nos2b transcripts were not robustly detected in cxcl18b⁺ MG clusters in our scRNA dataset. This discrepancy may be caused by technical limitations of scRNA-seq in capturing low-abundance or transient transcripts such as nos2b, as discussed in response to comment #11.

(13) The data in Figure 7 are interesting and suggest a link between NO signaling and notch activity. The use of the C-PTIO NO scavenger is not specific to MG, which limits the conclusions related to autocrine NO signaling in cxcl18b+ MG.

We acknowledge that the use of C-PTIO cannot distinguish between NO signaling within MG and paracrine effects from other retinal cells. Currently, technical limitations prevent MG-specific NO depletion. We have discussed this limitation accordingly in our revised “Limitations of this study” section (revised lines 540-545: “2. While our data suggest that injury-induced NO suppresses Notch signaling activation and promotes MG proliferation, the use of a general NO scavenger (C-PTIO) does not allow us to determine whether this regulation occurs in an autocrine or paracrine manner. The specific role of NO signaling within cxcl18b⁺ MG requires further validation using MG-specific NO depletion.”)

(14) Line 446-448. As mentioned before, the data do not support a causative link between microglia recruitment and cxcl18b induction. More specifically, dexamethasone is a broad-spectrum anti-inflammatory drug that blocks microglia activation and recruitment. Critically, the authors demonstrate that expression of cxcl18b occurs prior to microglia recruitment (see Figure 1, Supplement 1). Thus, the statement that cxcl18b induction depends on microglia recruitment is not accurate.

We thank the reviewer for reiterating this important point. We fully agree that the current data do not support a direct causal relationship between microglia recruitment and cxcl18b induction. As also addressed in our response to Comment 7, dexamethasone, as a broad-spectrum anti-inflammatory agent, cannot distinguish microglia-specific effects from those of other immune components. We have revised the text in revised lines 251–258 to clarify that microglia-mediated inflammation is associated with—but not required for—activation of cxcl18b-defined transitional MG states.

Reference:

Bogdan, C. (2001). Nitric oxide and the immune response. Nature immunology 2, 907-916.

Bradley, S., Tossell, K., Lockley, R., and McDearmid, J.R. (2010). Nitric oxide synthase regulates morphogenesis of zebrafish spinal cord motoneurons. The Journal of neuroscience : the official journal of the Society for Neuroscience 30, 16818-16831.

Gorsuch, R.A., Lahne, M., Yarka, C.E., Petravick, M.E., Li, J., and Hyde, D.R. (2017). Sox2 regulates Müller glia reprogramming and proliferation in the regenerating zebrafish retina via Lin28 and Ascl1a. Experimental eye research 161, 174-192.

Hamon, A., García-García, D., Ail, D., Bitard, J., Chesneau, A., Dalkara, D., Locker, M., Roger, J.E., and Perron, M. (2019). Linking YAP to Müller Glia Quiescence Exit in the Degenerative Retina. Cell reports 27, 1712-1725.e1716.

Iribarne, M., and Hyde, D.R. (2022). Different inflammation responses modulate Müller glia proliferation in the acute or chronically damaged zebrafish retina. Frontiers in cell and developmental biology 10, 892271.

Jin, D., Ni, T.T., Hou, J., Rellinger, E., and Zhong, T.P. (2009). Promoter analysis of ventricular myosin heavy chain (vmhc) in zebrafish embryos. Developmental dynamics : an official publication of the American Association of Anatomists 238, 1760-1767.

Krylov, A., Yu, S., Veen, K., Newton, A., Ye, A., Qin, H., He, J., and Jusuf, P.R. (2023). Heterogeneity in quiescent Müller glia in the uninjured zebrafish retina drive differential responses following photoreceptor ablation. Frontiers in molecular neuroscience 16, 1087136.

Lahne, M., Nagashima, M., Hyde, D.R., and Hitchcock, P.F. (2020). Reprogramming Müller Glia to Regenerate Retinal Neurons. Annual review of vision science 6, 171-193.

Lee, M.S., Jui, J., Sahu, A., and Goldman, D. (2024). Mycb and Mych stimulate Müller glial cell reprogramming and proliferation in the uninjured and injured zebrafish retina. Development (Cambridge, England) 151.

Lourenço, R., Brandão, A.S., Borbinha, J., Gorgulho, R., and Jacinto, A. (2021). Yap Regulates Müller Glia Reprogramming in Damaged Zebrafish Retinas. Frontiers in cell and developmental biology 9, 667796.

Matrone, G., Jung, S.Y., Choi, J.M., Jain, A., Leung, H.E., Rajapakshe, K., Coarfa, C., Rodor, J., Denvir, M.A., Baker, A.H., et al. (2021). Nuclear S-nitrosylation impacts tissue regeneration in zebrafish. Nat Commun 12, 6282.

Mazzolini, J., Le Clerc, S., Morisse, G., Coulonges, C., Kuil, L.E., van Ham, T.J., Zagury, J.F., and Sieger, D. (2020). Gene expression profiling reveals a conserved microglia signature in larval zebrafish. Glia 68, 298-315.

Meyers, J.R., Hu, L., Moses, A., Kaboli, K., Papandrea, A., and Raymond, P.A. (2012). β-catenin/Wnt signaling controls progenitor fate in the developing and regenerating zebrafish retina. Neural development 7, 30.

Nagashima, M., and Hitchcock, P.F. (2021). Inflammation Regulates the Multi-Step Process of Retinal Regeneration in Zebrafish. Cells 10.

Nathan, C., and Xie, Q.W. (1994). Nitric oxide synthases: roles, tolls, and controls. Cell 78, 915-918.

Pollak, J., Wilken, M.S., Ueki, Y., Cox, K.E., Sullivan, J.M., Taylor, R.J., Levine, E.M., and Reh, T.A. (2013). ASCL1 reprograms mouse Muller glia into neurogenic retinal progenitors. Development (Cambridge, England) 140, 2619-2631.

Rochon, E.R., Missinato, M.A., Xue, J., Tejero, J., Tsang, M., Gladwin, M.T., and Corti, P. (2020). Nitrite Improves Heart Regeneration in Zebrafish. Antioxidants & redox signaling 32, 363-377.

Sarich, S.C., Sreevidya, V.S., Udvadia, A.J., Svoboda, K.R., and Gutzman, J.H. (2025). The transcription factor Jun is necessary for optic nerve regeneration in larval zebrafish. PloS one 20, e0313534.

Sifuentes, C.J., Kim, J.W., Swaroop, A., and Raymond, P.A. (2016). Rapid, Dynamic Activation of Müller Glial Stem Cell Responses in Zebrafish. Investigative ophthalmology & visual science 57, 5148-5160.

Svahn, A.J., Graeber, M.B., Ellett, F., Lieschke, G.J., Rinkwitz, S., Bennett, M.R., and Becker, T.S. (2013). Development of ramified microglia from early macrophages in the zebrafish optic tectum. Developmental neurobiology 73, 60-71.

Thomas, D.D., Ridnour, L.A., Isenberg, J.S., Flores-Santana, W., Switzer, C.H., Donzelli, S., Hussain, P., Vecoli, C., Paolocci, N., Ambs, S., et al. (2008). The chemical biology of nitric oxide: implications in cellular signaling. Free radical biology & medicine 45, 18-31.

Thomas, J.L., Ranski, A.H., Morgan, G.W., and Thummel, R. (2016). Reactive gliosis in the adult zebrafish retina. Experimental eye research 143, 98-109.

Wan, J., and Goldman, D. (2016). Retina regeneration in zebrafish. Current opinion in genetics & development 40, 41-47.

White, D.T., Sengupta, S., Saxena, M.T., Xu, Q., Hanes, J., Ding, D., Ji, H., and Mumm, J.S. (2017). Immunomodulation-accelerated neuronal regeneration following selective rod photoreceptor cell ablation in the zebrafish retina. Proceedings of the National Academy of Sciences of the United States of America 114, E3719-e3728.

Yao, K., Qiu, S., Tian, L., Snider, W.D., Flannery, J.G., Schaffer, D.V., and Chen, B. (2016). Wnt Regulates Proliferation and Neurogenic Potential of Müller Glial Cells via a Lin28/let-7 miRNA-Dependent Pathway in Adult Mammalian Retinas. Cell reports 17, 165-178.

Yin, Z., Kang, J., Xu, H., Huo, S., and Xu, H. (2024). Recent progress of principal techniques used in the study of Müller glia reprogramming in mice. Cell regeneration (London, England) 13, 30.

Yu, C., Li, X., Ma, J., Liang, S., Zhao, Y., Li, Q., and Zhang, R. (2024). Spatiotemporal modulation of nitric oxide and Notch signaling by hemodynamic-responsive Trpv4 is essential for ventricle regeneration. Cellular and molecular life sciences : CMLS 81, 60.